# ON FINETUNING TABULAR FOUNDATION MODELS

## ABSTRACT

Foundation models are an emerging research direction in tabular deep learning. Notably, TabPFNv2 recently claimed superior performance over traditional GBDT-based methods on small-scale datasets using an in-context learning paradigm, which does not adapt model parameters to target datasets. However, the optimal finetuning approach for adapting tabular foundational models, and how this adaptation reshapes their internal mechanisms, remains underexplored. While prior works studied finetuning for earlier foundational models, inconsistent findings and TabPFNv2's unique architecture necessitate fresh investigation. To address these questions, we first systematically evaluate various finetuning strategies on diverse datasets. Our findings establish full finetuning as the most practical solution for TabPFNv2 in terms of time-efficiency and effectiveness. We then investigate how finetuning alters TabPFNv2's inner mechanisms, drawing an analogy to retrieval-augmented models. We reveal that the success of finetuning stems from the fact that after gradient-based adaptation, the dot products of the query-representations of test objects and the key-representations of in-context training objects more accurately reflect their target similarity. This improved similarity allows finetuned TabPFNv2 to better approximate target dependency by appropriately weighting relevant in-context samples, improving the retrieval-based prediction logic. From the practical perspective, we managed to finetune TabPFNv2 on datasets with up to 50K objects, observing performance improvements on almost all tasks. More precisely, on academic datasets with I.I.D. splits, finetuning allows TabPFNv2 to achieve state-of-the-art results, while on datasets with gradual temporal shifts and rich feature sets, TabPFNv2 is less stable and prior methods remain better.

## 1 INTRODUCTION

Recently, deep learning for tabular data has rapidly advanced (Gorishniy et al., 2021; Somepalli et al., 2021; Gorishniy et al., 2024; Hollmann et al., 2023; Ye et al., 2024; Holzmüller et al., 2024), frequently drawing inspiration from natural language processing (NLP) and computer vision (CV), where foundational models — large-scale architectures pretrained on vast datasets and adaptable to diverse tasks (Radford et al., 2021; Ramesh et al., 2021; Alayrac et al., 2022) — have become pivotal for sample-efficient learning. The application of such models to the tabular domain was initially uncertain due to its inherent heterogeneity and the scarcity of large, public pretraining datasets. However, TabPFN (Hollmann et al., 2023) demonstrated their potential by pioneering pretraining on diverse synthetic datasets designed to mimic real-world distributions. Its recent successor, TabPFNv2 (Hollmann et al., 2025), further validated this approach, showing its synthetically learned priors enable it to outperform leading GBDT implementations (Prokhorenkova et al., 2018; Ke et al., 2017; Chen & Guestrin, 2016) on small tabular datasets.

While TabPFNv2's superiority over GBDTs was demonstrated using in-context learning — where the entire training set serves as its input prompt — it is not entirely clear how more computationally intensive gradient-based adaptations, such as full/partial finetuning or parameter-efficient methods like LoRA (Hu et al., 2021), might affect its performance. This uncertainty is particularly noteworthy because, while common sense intuition implies that finetuning is universally beneficial, the recent NLP findings report that pure in-context learning can sometimes outperform finetuning (Yin et al., 2024). Although several recent works (Feuer et al., 2024; Thomas et al., 2024; Ma et al., 2024; Xu et al., 2024; den Breejen et al., 2023) have finetuned tabular foundational models, these efforts were often not systematic, formed part of larger pipelines, largely focused on the outdated TabPFN model, and crucially, did not analyze how finetuning alters the internal mechanisms of foundational models.

The main focus of our work is to understand how finetuning impacts the inner logic of TabPFNv2. To identify the optimal finetuning regime for this in-depth analysis, we first systematically compared various strategies on a diverse set of datasets with up to $\approx$1M total cells (columns $\times$ rows) [1]. Contrary to previous works (Xu et al., 2024; Feuer et al., 2024) that advocate for partial model adaptation to prevent overfitting, our findings indicate that full finetuning, when properly configured (including hyperparameter ablation for efficient and stable adaptation), appears to be the superior option compared to partial and parameter-efficient alternatives. This result led us to select full finetuning as our chosen method for detailed investigation.

Our subsequent analysis draws parallels between retrieval-based tabular models and TabPFNv2. Within TabPFNv2's last layer, the dot products between query-representations of test objects and key-representations of in-context samples provide signals used by attention to weight training examples. We find that after task-specific finetuning, these query-key dot products exhibit a significantly stronger alignment with actual target similarity. This more precise correspondence, a direct result of finetuning, greatly simplifies the problem for attention, which can more effectively approximate the test label by precisely weighting the most relevant in-context samples. In particular, we observe that the majority of finetuning performance gains come from samples where inter-sample attention becomes more sharply concentrated after finetuning.

Finally, we put the TabPFNv2 model and its finetunes in the modern tabular DL context and compare it to the recent SoTA models (Ye et al., 2024; Gorishniy et al., 2025) and additionally evaluate it on a new challenging benchmark (Rubachev et al., 2025).

To sum up, our contributions are the following:

1. We extensively compare different finetuning regimes for the TabPFNv2 model and establish simple full finetuning as a strong and stable baseline for TabPFNv2 adaptation, contrary to prior work (Feuer et al., 2024; Xu et al., 2024) where the necessity of partial finetuning methods was emphasized for preventing overfitting.

2. We analyse the finetuning's impact drawing a parallel between TabPFNv2 and retrieval-based models. We demonstrate that finetuning refines TabPFNv2 by ensuring the dot products of query-representations (test object tokens in inter-sample attention) and key-representations (in-context samples) more accurately reflect target similarity. This improved alignment simplifies the prediction problem for the model, enabling it to more effectively approximate the target based on in-context examples.

3. We provide a thorough comparison of original and finetuned TabPFNv2 against the state-of-the-art tabular deep learning methods. Our analysis includes datasets with up to 1M cells (rows $\times$ columns) – reflecting the current computational limit for the straightforward finetuning – and spans both traditional academic benchmarks and more challenging real-world datasets with temporal shifts and rich feature sets. On academic benchmarks, we find that non-finetuned TabPFNv2 performs on par with strong MLP-PLR baseline and the finetuned version achieves state-of-the-art results. Conversely, on more challenging real-world datasets, both TabPFNv2 and its finetunes often perform less stable compared to non-foundational DL methods.

## 2 RELATED WORK

Here we briefly outline research lines relevant to our study.

**Tabular Deep Learning**. In recent years, deep learning models for tabular data have emerged as strong contenders to traditional "shallow" methods like Gradient Boosting Decision Trees (GBDTs). Indeed, recent DL models (Gorishniy et al., 2024; Holzmüller et al., 2024; Ye et al., 2024; Gorishniy et al., 2025; Hollmann et al., 2025) have often matched or surpassed leading GBDT implementations (Prokhorenkova et al., 2018; Ke et al., 2017; Chen & Guestrin, 2016). This progress stems from innovations in architectures (Gorishniy et al., 2021; Somepalli et al., 2021), regularizations (Jeffares et al., 2023), and learning protocols (Holzmüller et al., 2024; Bahri et al., 2021; Rubachev et al., 2022),

---

[1] We use datasets generally larger than those in the TabPFNv2 paper (Hollmann et al., 2025), but only those where finetuning on a single 80GB GPU is possible.

leveraging the models capabilities not readily available to GBDTs. Our work explores foundational models, a paradigm that is also inherently applicable only for deep learning models.

**Foundational Models** currently dominate among deep learning solutions for CV and NLP problems and have become a key component in the most state-of-the-art systems Radford et al. (2021); Alayrac et al. (2022). To put simply, foundational models are the models pretrained on vast amounts of available data from some domain that can then be adapted to a wide number of downstream tasks from this domain. The knowledge captured during pretraining often acts as a valuable prior, which is particularly beneficial in few-shot learning scenarios.

Adapting foundational models to specific tasks typically follows one of two main pathways. The first, finetuning, involves further training the model on the downstream dataset to optimize its parameters for the new task. For enhanced sample and runtime efficiency, this often includes updating only a subset of parameters, such as specific layers, low-rank adapters (LoRA) (Hu et al., 2021), or learnable prompts (Lester et al., 2021). In contrast, in-context learning adapts the model without altering its parameters, instead providing downstream training samples as part of the input prompt or context (Brown et al., 2020). The choice between finetuning and in-context learning often depends on factors like downstream dataset size, computational resources, and the need for multi-task adaptation. While finetuning is often considered more effective (Liu et al., 2022), recent studies suggest in-context learning can be competitive or even preferable in certain setups (Yin et al., 2024).

**TabPFN** (Hollmann et al., 2023), the pioneering foundational model for tabular data, was designed to address a wide array of tabular tasks off-the-shelf. It employs a transformer-like architecture and utilizes in-context learning, with the entire downstream training set serving as its prompt. Its pretraining relies on numerous synthetic datasets engineered to mirror common application-specific tabular tasks. The successor, TabPFNv2 (Hollmann et al., 2025), enhances this with a more powerful architecture, pretraining on a broader spectrum of synthetic data, and sophisticated feature and target preprocessing, demonstrating superior performance over GBDTs, especially on small-scale datasets. Our work builds upon this by systematically investigating gradient-based finetuning specifically for TabPFNv2. While some recent studies (Feuer et al., 2024; Thomas et al., 2024; Ma et al., 2024; Xu et al., 2024; den Breejen et al., 2023) have explored aspects of finetuning tabular foundational models, these explorations were often secondary to their main objectives. Furthermore, these studies predominantly used the original TabPFN model, which differs significantly from TabPFNv2 in capability and architecture, meaning their conclusions might not directly apply. For example, Feuer et al. (2024) noted potential overfitting with full finetuning of TabPFN on validation sets, a phenomenon we did not encounter in our TabPFNv2 experiments. Importantly, the original paper (Hollmann et al., 2025) benchmarks TabPFNv2 primarily against GBDT-based baselines. To obtain a broader understanding of TabPFNv2 relative performance, we thoroughly compare it to existing non-foundational tabular DL models and show that it is specifically finetuning that enables TabPFNv2 to achieve the state-of-the-art performance.

## 3    REVISITING FINETUNING STRATEGIES FOR TABPFNV2

In this section, we systematically evaluate different finetuning techniques for TabPFNv2. Through this evaluation we aim to establish a strong finetuning baseline for the TabPFNv2 model to address the lack of consensus or information (in case of the second version) on best TabPFN finetuning methodology in literature.

**Evaluation Protocol**. We experiment with finetuning TabPFNv2 on two established tabular DL benchmarks from (Grinsztajn et al., 2022) and (Gorishniy et al., 2024). We use only the datasets for which it is possible to finetune TabPFNv2 using the entire dataset as an input prompt on a single GPU with 80GB of memory. We provide a list of datasets we have used with their characteristics in Table 6. Our benchmark covers larger datasets than those used for evaluation in Hollmann et al. (2025) – the average dataset size used in our paper is approximately 15K examples, while the average dataset size in Hollmann et al. (2025) is at approximately 3K. This potentially presents a more challenging setting for the TabPFNv2 model and it also allows us to compare the foundational model to strong non-foundational tabular DL baselines, while in Hollmann et al. (2025) comparison is limited to GBDTs and simple baselines like SVMs.

For all TabPFNv2 finetuning runs and ablations, we tune the learning rate on the validation set using the logspace grid with 10 learning rate values: `logspace(5e-6, 5e-4)`. For other baselines, we use hyperparameter grids from (Gorishniy et al., 2025) and tune for 100 iterations. We always use 1024 objects to calculate the loss per gradient step (except the batch size ablations below), while the rest objects are used as an input prompt. For early stopping, we compute the performance on the validation subset every ten gradient steps and stop the finetuning after 16 non-improving evaluations. We report the RMSE and classification accuracy for regression and classification problems, respectively. Additionally, we use the relative improvement to the MLP baseline metric introduced in Gorishniy et al. (2025) (the $R^2$ and accuracy are used to compute the relative improvement to the tuned MLP configuration). For additionall details regarding the experimental protocol refer to the Appendix A. Below we provide a brief overview of the finetuning strategies we are re-evaluating for TabPFNv2.

**Full finetuning** is the most straightforward way to adapt pretrained models, used in other domains like NLP (Howard & Ruder, 2018; He et al., 2015) and computer vision (Kolesnikov et al., 2020; Beyer et al., 2024). But there is no consensus in prior work on finetuning tabular foundation models (Feuer et al., 2024; den Breejen et al., 2023) – albeit, based on TabPFNv1.

**Parameter-efficient finetuning (PEFT)** is popular for LLM adaptation. While with the current tabular foundation model scale (7M parameters) the memory efficiency gains from PEFT are not very important, the inductive biases and potential implicit regularization of partial finetuning might be beneficial to prevent overfitting, as previously hypothesized in Feuer et al. (2024). We consider the following options for parameter-efficient finetuning:

- **Low Rank Adapters (LoRA)** – Hu et al. (2021) is a widely used parameter-efficient finetuning method that uses two low-rank matrices to update a pretrained full-rank matrix.
- **Last layers** – finetuning only the upper layer is also a popular partial finetuning method, that is occasionally used for finetuning pretrained models in other modalities (Hu et al., 2021; Lee et al., 2019).
- **LayerNorm, Head and Embeddings** – finetuning only the feature and target linear embedding layers, MLP prediction head and the affine layer normalization parameters. These parameters represent a small fraction of the whole model parameters, but have been found important in the model adaptation literature (Zhao et al., 2024; Chen et al., 2024).
- **Numerical Feature Embeddings** are a popular and useful technique in the recent tabular DL architectures (Gorishniy et al., 2022; 2025; Holzmüller et al., 2024), which is not yet exploited in tabular foundation models (especially its interaction with TabPFNv2). We experiment with adding the embedding modules before the main TabPFNv2 backbone and finetuning them together with the model.

**Practical observations about finetuning & Training Time**. Before comparing all the performance results we highlight some practical observations. First, we measure the training times of different finetuning methods. We can see in Table 1 that *full finetuning is the most efficient finetuning method in terms of convergence speed*. Second, *we demonstrate that using larger batches during finetuning improves the finetuning performance*. In more details, we keep the training context size fixed and increase the prediction sequence length. From results in Table 2 we can see that larger batches result in the higher performance of finetuned models. Furthermore, we provide a discussion on computational complexity related to finetuning the TabPFNv2 model in Appendix D.

Table 1: Comparison of time in seconds spent on finetuning (FT.) and in-context prediction (No FT.).

|  | CA ↓ | HO ↓ | DI ↓ | CH ↑ | AD ↑ |
|---|---|---|---|---|---|
| LoRA | 328 | 1330 | 3607 | 105 | 4155 |
| Emb.,LN, Head | 390 | 766 | 995 | 122 | 1978 |
| Full | 155 | 468 | 1777 | 74 | 1480 |
| No FT. | 12 | 24 | 36 | 1 | 18 |

Table 2: The effect of the number of objects used in prediction during training. We can see that using more objects to make one gradient estimation is beneficial.

| Pred. Length | CA ↓ | HO ↓ | DI ↓ | CH ↑ | AD ↑ |
|---|---|---|---|---|---|
| 2 | 0.3963 | 3.0755 | 0.1332 | 0.8535 | 0.8588 |
| 128 | 0.3839 | **2.9930** | 0.1303 | 0.8488 | 0.8680 |
| 1024 | **0.3822** | **2.9919** | **0.1275** | **0.8647** | **0.8710** |

**Optimal finetuning strategy**. We summarize the results in Figure 1, the full results are available in Table F. We can see that *fine-tuning in general makes a significant impact on model performance compared to pure in-context performance*. However, the difference between full finetuning and all

considered PEFT variations is minimal. This observation and training efficiency results in Table 1 make *full finetuning* a *go-to simple baseline for TabPFNv2 adaptation*.

Furthermore, introducing untied (non-shared) linear or advanced piecewise-linear embeddings has marginal effect on finetuning performance and indicates that either the TabPFNv2 model has already learned the feature transforms and advanced embeddings are not needed or there needs to be a more sophisticated embedding scheme (e.g. during pretraining), which is an interesting exploration direction for future work.

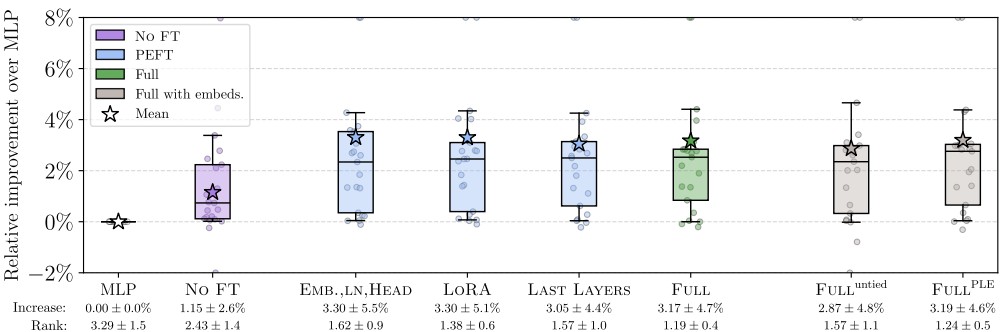

Figure 1: Comparison of different TabPFNv2 finetuning methods. The plot summarizes the relative performance improvement over a tuned MLP baseline. We consider off-the-shelf **No FT** TabPFNv2 with no finetuning compared to parameter efficient tuning methods **PEFT**, full model finetuning **Full** and finetuning with modified **Feature Embeddings**. Box plots summarize the results on all datasets from Table 6, bars represent the 25th, 50th and 75th percentiles, whiskers represent the 10th and 90th percentiles (visualization from Gorishniy et al. (2025)). The numbers at the bottom of the plot present average improvement over MLP and average ranks.

### HOW DOES FINETUNING COMPARE TO TRAINING FROM RANDOM INITIALIZATION?

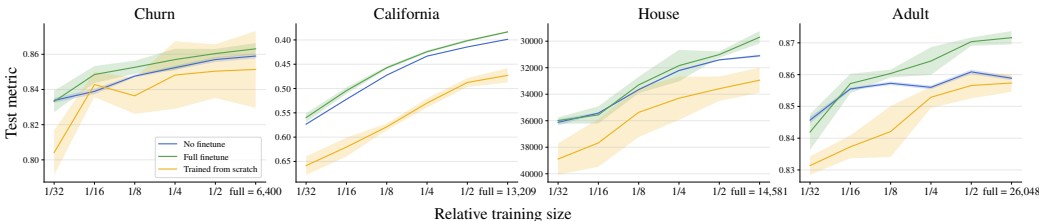

Figure 2: Performance of different methods on subsamples of four datasets. Churn and Adult are binary classification datasets and the metric is accuracy (higher is better), California and House are regression datasets with the RMSE test metric (lower is better, axis flipped). The shaded area represents 2 standard deviations in both directions from the average accuracy.

When we can afford finetuning a TabPFNv2 model – we can also pretrain the model from scratch on a target dataset. In this section, we study how pretraining on around 130M synthetic datasets used for TabPFNv2 compares to training from scratch on a target dataset depending on its size. Furthermore we study how the benefits that come from finetuning change with varying dataset sizes.

To answer these questions, we perform the following experiments. First, for four datasets (Churn, California, House, Adult), we create their subsamplings, with a train set in each following subsampling being half of the previous one, while validation and test sets are the same. We then compare the performances of the finetuned TabPFNv2 model, original TabPFNv2 model, and the model with the TabPFNv2 architecture trained from scratch on each of the subsampled versions of each dataset.

Figure 2 shows that *finetuning TabPFNv2 provides more benefits for larger target datasets* (clearly seen on House and Adult), while for smaller datasets the performance improvements from finetuning are often statistically insignificant compared to pure in-context learning.

Furthermore, while on some datasets, such as Adult and Churn, training from scratch comes close to the performance of the pretrained TabPFNv2 (especially for large problem sizes), there are datasets (e.g. California), where for all problem sizes pretraining gives a significant boost to performance, which is not achieved with training from scratch, but is further amplified via finetuning. This variance in the performance of pre-trained model and efficacy of finetuning warrants a deeper investigation into the inner workings of TabPFNv2 and its finetunes, which we describe in the following section.

# 4 DISSECTING FINETUNING'S IMPACT ON TABPFNV2

As shown in the previous section, finetuning significantly enhances TabPFNv2 performance on our testbed of medium-scale datasets. In this section we aim to understand how and to what extent these improvements are achieved by examining finetuning influence on TabPFNv2 internal behavior, particularly its (inter-sample) attention mechanism.

Our analysis is based on the intuition that pretrained TabPFNv2 functions similarly to retrieval-augmented models like ModernNCA (Ye et al., 2024) or TabR (Gorishniy et al., 2024) when making predictions, this contrasts with alternative explanations of in-context learning in other modalities through e.g. performing implicit SGD (Von Oswald et al., 2023), or implementing complex algorithmic circuits (Olsson et al., 2022; Nanda et al., 2023).

Specifically, we conjecture that an important part of TabPFNv2 prediction mechanics is an implicit retrieval mechanism implemented via inter-sample attention over train dataset objects (in particular, the labels) – resembling retreival-based models, which perform this explicitly. The first evidence supporting this analogy comes from comparing performance gains over an MLP baseline achieved by the in-context TabPFNv2, ModernNCA, and MLP-PLR across all datasets in our benchmark. Improvements from TabPFNv2 showed a high correlation (**0.89** Pearson correlation) with improvements from ModernNCA, while a similar correlation for the identically per-

Table 3: Comparison of weighted kNN prediction (or MNCA-like prediction) with attention scores from last layer of TabPFNv2 as a proxy of similarity between instances. Attention weights after finetuning better reflect similarity that results in more accurate predictions. The test scores are calculated on a single seed.

|  | CA ↓ | HO ↓ | DI ↓ | CH ↑ | AD ↑ | pol ↓ |
|---|---|---|---|---|---|---|
| **TabPFNv2 performance**: | | | | | | |
| No FT | 0.398 | 3.105 | 0.137 | 0.859 | 0.859 | 4.830 |
| Finetuned | 0.382 | 2.960 | 0.127 | 0.867 | 0.872 | 2.801 |
| **TabPFNv2 attn. weighted kNN performance**: | | | | | | |
| kNN $_{\text{via No FT}}$ | 0.473 | 3.404 | 0.183 | 0.856 | 0.856 | 6.845 |
| kNN $_{\text{via Finetune}}$ | 0.407 | 3.313 | 0.155 | 0.866 | 0.872 | 3.165 |

formant MLP-PLR model is more mild (**0.53** Pearson correlation). These results support characterizing TabPFNv2 as an advanced implicit retriever, guiding our subsequent investigation of finetuning.

Based on this intuition, we hypothesize that a primary effect of finetuning is the refinement of the similarity signals that TabPFNv2 uses to weight in-context examples. Specifically, we propose that finetuning improves how accurately the dot products between the query-representations (derived from test objects) and key-representations (derived from in-context training examples) in the model's final layer reflect the true target similarity between these objects. A more accurate underlying similarity measure would allow the attention softmax to more effectively identify and upweight the most informative training examples for predicting the test object's label.

To verify this hypothesis, we designed an experiment to directly assess the quality of the attention scores as proxies for target relevance. For both the original and the finetuned TabPFNv2 models, we extracted the attention scores assigned by the final layer's attention mechanism to each in-context training example for a given test instance. These attention weights were then used to compute a weighted average of the corresponding training example targets. The intuition behind this experiment is straightforward: if the attention scores accurately capture the relevance of training examples to a test example's target, then a weighted average of training targets using these scores should yield a good approximation of the test target. Thus, higher quality attention weights, better reflecting target similarity, should result in a more accurate target prediction. The results of this analysis, presented in Table 3, demonstrate a marked improvement: the weighted average of targets computed using attention scores from the finetuned TabPFNv2 aligns significantly more closely with the groundtruth test targets compared to those from the original TabPFNv2. This finding strongly supports our

hypothesis that finetuning refines the query-key dot products to better reflect true target closeness, thereby simplifying the task for the attention mechanism and improving the hypothesized "implicit retrieval".

Interestingly, the effect described above also affects the attention distribution itself. As shown by the histograms in Figure 3, for most datasets, finetuning leads to a notable decrease in the entropy of the attention distribution over the in-context training examples for a significantly large subset of test instances. This indicates that for these instances the finetuned model becomes "more focused", concentrating its attention mass on a smaller subset of training examples. This behavior is consistent with our earlier finding: if the underlying query-key dot products provide a clearer signal of which examples are truly most similar in terms of their targets, the model can confidently allocate higher attention to these few, most relevant neighbors, rather than spreading its attention more diffusely.

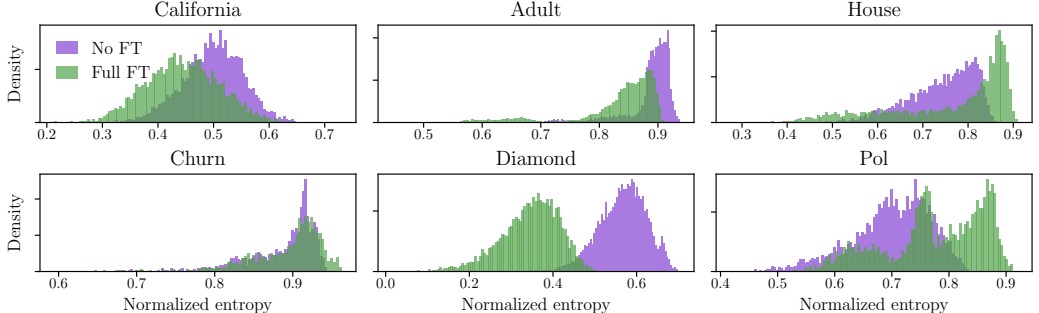

Figure 3: Normalized entropy of the attention weights from the last layer of TabPFNv2 (on test samples). On California, Adult and Diamond, attention weights consistently become more concentrated after finetuning. On Churn dataset, there is no notable shift, and high entropy indicates smooth distribution of weights. On House and Pol datasets, for almost 70-80% of the samples entropy increased but on Figure 4 we show that the most performance gains are obtained from those samples where entropy dropped. More detailed explaination is provided in Appendix C.

Finally, to connect these changes in the attention behaviour directly to predictive performance, we examined the relationship between the sample-wise change in the attention entropy (occured due to finetuning) and the corresponding sample-wise change in a chosen performance metric. Figure 4 plots this dependency.[2]

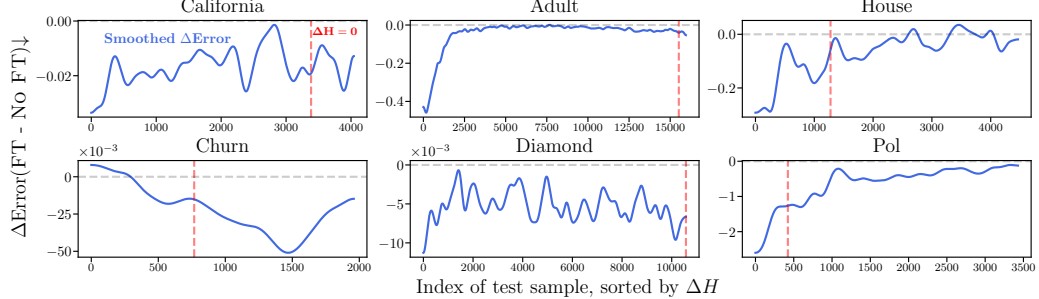

Figure 4: Dependency between the sample-wise change in prediction error (finetuned vs. original TabPFNv2) and the corresponding change in entropy of attention weights. In most cases, finetuned model considerably improves error on samples where entropy dropped significantly (indices closer to zero), i.e. where attention weights became more concentrated. The details are provided in Appendix C.

The results reveal an apparent regularity: the test datapoints contributing most substantially to the overall performance improvement (in other words, corresponding to the most negative $\Delta Error$ values) are those for which attention entropy decreased. Conversely, while some test datapoints do

---

[2]The details of plotting these graphs are presented in Appendix C

exhibit an increase in attention entropy after finetuning, their collective contribution to the overall performance change is substantially smaller. This nuanced observation reinforces the idea that the primary driver of finetuning's benefit lies in its ability to sharpen the model's focus on the most relevant in-context examples by improving the underlying similarity representations that guide the attention mechanism.

## 5 TABPFNV2 AND ITS FINETUNES IN A BROADER TABULAR DL CONTEXT

In this section, we compare the original TabPFNv2 model and its finetuned variant to the non-foundational state-of-the-art tabular DL models. Additionally, we evaluate TabPFNv2 on a subsampled versions of the TabReD benchmark (Rubachev et al., 2025) datasets.[3]

The original TabPFNv2 paper (Hollmann et al., 2025) reports the state-of-the-art results on small datasets and compares to classical "shallow" models and AutoML solutions. We aim to extend the scope of the experiments to bigger datasets and new dataset characteristics (like temporal shift and extensive feature engineering in TabReD) and compare to contemporary tabular DL methods that achieve the state-of-the-art performance in this setting (Ye et al., 2024; Gorishniy et al., 2025).

### 5.1 COMPARISON WITH SoTA TABULAR DL BASELINES

To evaluate TabPFNv2, we use the original in-context version and the in-context ensemble version (ensembled over different input and target preprocessing and transformations). Furthermore, we evaluate finetuned TabPFNv2, and its deep ensemble variant. We use five finetuning runs to construct the ensemble.

We compare TabPFNv2 and its finetuned versions to the following DL methods and classical ML baselines:

- **MLP$^{PLR}$** – MLP with periodic numerical feature embeddings (Gorishniy et al., 2022).
- **MNCA**: ModernNCA – a state-of-the-art non-parametric tabular DL model (Ye et al., 2024).
- **TabM$^{\dagger}_{mini}$** — A recent state-of-the-art parametric tabular DL model. We evaluate a variant that uses piece-wise linear numerical feature embeddings (denoted by †).
- **XGBoost**: In addition to deep models, we use a tuned GBDT model as a commonly accepted strong "shallow" baseline.

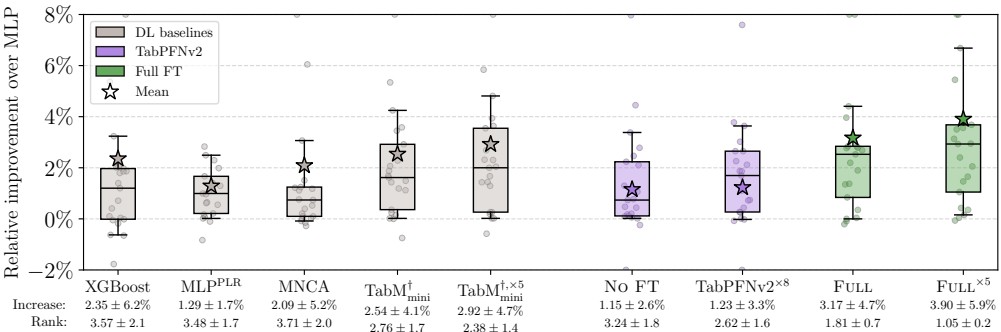

Figure 5: Comparison of TabPFNv2 (with and without finetuning) with other state-of-the-art tabular DL methods. The plot summarizes the relative performance improvement over a tuned MLP baseline. Box plots summarize the results on all datasets from Table 6. Notation follows Figure 1.

**Results summary**. Results of the comparison are provided in Figure 5. Below, we highlight our key observations. The original *TabPFNv2 used in the in-context learning regime and its ensemble variations are generally inferior to the up-to-date strong tabular DL models on academic datasets with up to 1M table cells* (rows × columns). Nevertheless, its performance is close to MLP-PLR, which is a strong baseline.

---

[3]Subsampling is done due to computational and engineering constraints

*Finetuning TabPFNv2 results in consistent improvements*. Furthermore, ensembling the finetuned TabPFNv2 model produces substantial performance improvements on top of single finetuned model, elaborating on efficient ensembling may be an interesting future research direction.

Overall, finetuned TabPFNv2 is a SoTA model on academic datasets where finetuning is computationally feasible on a single GPU. However it has inherent limitations in scalability due to finetuning and its similarity to retrieval-based models. We expand upon these limitations in the following subsection.

## 5.2 EVALUATION ON TABRED SUBSAMPLES

We provide the results of evaluating $TabM^{\dagger}_{mini}$, MNCA, TabPFNv2 and finetuned TabPFNv2 in Figure 6. We can see that on this version of the TabReD datasets the TabPFNv2 without finetuning is less stable compared to the current SoTA model "TabM". On almost all datasets finetuning does improve upon the in-context version (except Sberbank Housing where it often degrades performance).

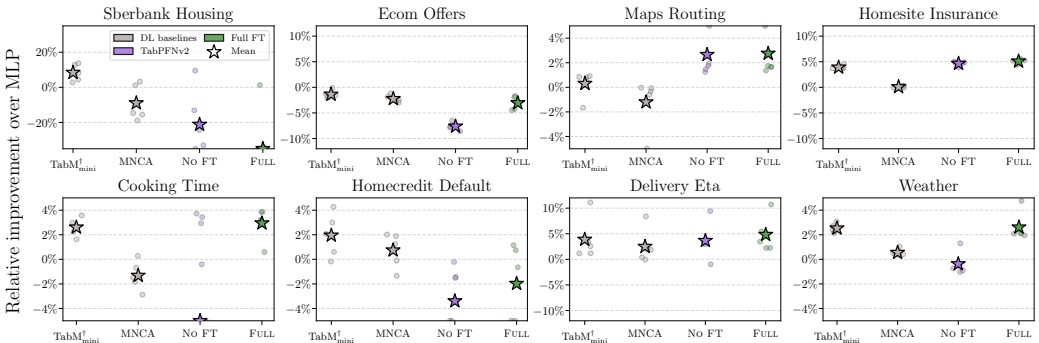

Figure 6: Results on five different TabReD dataset subsamples. We report average improvement over the tuned MLP baseline across five different subsamples for each dataset. For each subsample we average the score over five random seeds.

Overall, TabPFNv2 results on TabReD subsamples are less stable compared to the results on the academic benchmarks from previous sections – this may be related to the presence of temporal shift in these datasets which may pose challenges for models implementing retrieval-based predictions. We further discuss this in Appendix E. Furthermore, scaling finetuning to the full-sized datasets requires significant engineering efforts.

## 6 LIMITATIONS

**Choice of datasets.** In our experiments we use only the datasets that can fully be handled by the TabPFNv2 model on a single 80 GB GPU. Therefore, our conclusions should be additionally verified for the large-scale downstream problems, where data has to be fed to TabPFNv2 by chunks and the finetuning procedure should be sufficiently altered.

**Different feature and target preprocessing.** In the comparison to the non-foundational tabular DL models, we did not standardize the data/target preprocessing across TabPFNv2 and other methods. We assumed that the preprocessing recommendations provided by the authors of each method were close to optimal and decided to use them as prescribed in the original papers.

## 7 CONCLUSION

In this paper, we systematically investigated gradient-based adaptation for TabPFNv2. Our findings establish full finetuning as the optimal strategy, crucially revealing its success stems from refining internal similarity assessments for improved retrieval-based prediction. While it elevates finetuned TabPFNv2 to state-of-the-art performance on academic datasets where finetuning is technically feasible (currently there is a limit in dataset size), significant challenges persist regarding scalability and robustness to real-world data complexity like shifts or complex real-world features. Future research should therefore prioritize developing scalable adaptation methods and enhancing resilience to the complexities of diverse tabular data.

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

## A    REPRODUCIBILITY STATEMENT

We provide the code for TabPFNv2 finetuning in the supplementary material as an archive with the source code. To run it one must also obtain the TabPFNv2 checkpoints from hf.co/Prior-Labs. We reuse the datasets from (Gorishniy et al., 2024). See the `bin/tabpfnv2_finetune.py` script and `exp/full-finetune/adult/evaluation/0.toml` config file for an entry point.

## B    THE USE OF LARGE LANGUAGE MODELS

Our use of LLMs is limited to writing aid and basic controlled coding assistance (e.g. text stylistic improvements, grammar checking, code for polishing figures and/or tables).

## C    DETAILS OF ANALYSIS OF THE TABPFNv2 FINETUNING

First, we briefly explain the methodology used to obtain Table 3. Then, we expand on the technical details of generating the results shown in Figure 3 and Figure 4. Finally, we provide an extended discussion and interpretation of these figures.

**kNN Score Calculation in Table 3.** To understand how finetuning affects the ability of TabPFNv2 to reflect target similarity, we employ a retrieval-like prediction mechanism based on attention scores. For each test sample, we extract attention scores from the last layer of TabPFNv2. Since attention is calculated between the test sample and each train sample, attention scores form a similarity distribution over train samples. These attention scores $\left(w \in \mathbb{R}^{N_{train}}\right)$ are then used for each test sample to make a weighted average prediction over train set, i.e. for regression, $\hat{y} = \sum_{i}^{N_{train}} y_i w_i$ and, for classification, we sum the logits of the corresponding class of train neighbors.

**Normalized Entropy Calculation.** The normalized entropy is computed using the same last-layer attention scores from TabPFNv2. For each test sample we calculate the entropy of the score distribution over train samples and normalize it dividing by $\ln(\text{train size})$. Figure 3 shows the distribution of this normalized entropy across the test set for six different datasets.

**Generation of Figure 4.** For each test sample, we calculate the error (MSE/LogLoss) and the normalized attention entropy for both the finetuned (FT) and the off-the-shelf TabPFNv2 (No FT).

Test samples are then sorted based on the change in normalized entropy in ascending order. The vertical axis of the plots the change in error for each sample. So, y-axis shows the change in error between 'FT' and 'No FT' (lower difference means FT is better), and x-axis shows an index of test sample, with lower indices corresponding to larger decreases in entropy. The red line shows the index where entropy change becomes positive ($\Delta H = 0$). The blue line shows the change in error after smoothing with a Gaussian filter to illustrate the overall trend.

**Interpretation and Discussion.** Overall, these three artifacts collectively support the hypothesis that a primary effect of finetuning is the refinement of the similarity signals that TabPFNv2 uses to weight in-context examples:

- Table 3 shows that for all datasets attention scores after finetuning more accurately reflect similarity of targets *on average* across test samples. However, Table 3 does not specify which individual samples benefit most from finetuning.
- Figure 3 shows the change in distribution of attention scores. For California, Adult and Diamond the distribution of the attention scores becomes consistently more concetrated – for 80-100% of the test samples entropy decreased. We believe that this concentration is a primary driver of finetuning benefits. Lower entropy suggests that higher attention weights are assigned to the "closest" neighbors, indicating improved latent space for calculating similarity between objects – since the metrics in Table 3 improved. However, the patterns on House and Pol are more nuanced, necessitating a deeper analysis via Figure 4.
- Figure 4 reveals that for all datasets, except Churn, the error improved the most on those test samples where entopy decreased the most. In other words, the test datapoints contributing most substantially to the overall performance improvement (lowest $\Delta Error$) are those for which attention entropy decreased the most ($\Delta H < 0$). Even for House and Pol the error clearly improves on samples where entropy decreased but this improvement decays on samples where entropy increased (indices to the right of the red line).
- The results on Churn dataset are not fully aligned with all our findings, we keep them for transparency. Although entropy distribution remains unchanged and the trend line in Figure 4 does not reflect the effects described above, the attention weights still improve after finetuning as can be seen by results in Table 3. This observation motivates deeper investigation into the inner workings of the model, and the particular mechanisms which improve the latent space for calculating similarity between objects. A more thorough analysis of such cases is reserved for future work.

## D  EFFICIENCY DISCUSSION

Here we discuss the efficiency aspects that are relevant for tabular foundation models sharing the TabPFNv2 architecture and especially noticable on bigger datasets. We provide per-epoch timings and relevant dataset dimensions in Table 4.

As described in the main paper, we limit all experiments to 1 M cells ($N \cdot M \leq 10^6$, where $N$ is the number of rows and $M$ is the number of features). TabPFNv2 uses attention over rows and attention over features, so the total number of attention operations scales as $O\big(NMd\,(N + M + 2d)\big)$, with $d$ denoting the query/key/value dimension. So, the DI dataset — which incurs the most operations—is the slowest. We also include the Cooking and Homesite datasets, which have many features but few samples. While these datasets involve fewer operations than DI, their per-epoch time is high because gradient checkpointing is enabled only for these two datasets (due to high memory demands due to large intermidiate activations required for backprop in finetuning). Overall, the hard limit for finetuning on one GPU without resorting to parallelism is at 1M cells (because of activation memory) and time-wise performance is summarized by the number of attention operations which is $O\big(NMd\,(N + M + 2d)\big)$.

## E  ANALYSIS OF THE FINETUNING FAILURES

In this section we look into the few failure cases of the TabPFNv2 finetuning. There are two datasets (sberbank-housing subsample and KDDCup09_upselling) where finetuning actually degrades performance of the base TabPFNv2. We link this to overfitting as explained below.

Table 4: Dataset Properties, Computational Requirements and the actual time per epoch

| Property/Dataset | CH | CA | HO | AD | DI | Cooking Subsample | Homesite Subsample |
|---|---|---|---|---|---|---|---|
| # Rows ($N$) | 6400 | 13 209 | 14 581 | 26 048 | 34 521 | 5208 | 3344 |
| # Features ($M$) | 11 | 8 | 16 | 14 | 9 | 190 | 299 |
| # Attention ops. | 9.2e10 | 2.8e11 | 6.7e11 | 1.9e12 | 2.1e12 | 1.1e12 | 7.7e11 |
| Time per epoch | 3s | 8s | 16s | 39s | 46s | 41s | 34s |

Both datasets have extreme feature-to-sample ratios, with KDD Upselling having the highest ratio among our 21 academic datasets. The sberbank-housing dataset additionally possesses a temporal shift (Rubachev et al., 2025) which may make it even more prone to overfitting during finetuning.

To investigate temporal shift specifically, we evaluated sberbank-housing on temporal vs. random splits (5 splits, 5 seeds each). The results in Table 5 demonstrate that temporal shifts causes performance degradation. Fine-tuned TabPFNv2 achieves best performance on random splits but suffers under temporal distribution shift, while TabM excels on temporal splits.

Table 5: Model Performance Comparison Across Different Splits

| Split/Model | MLP | MNCA | TabM_mini | No FT | Full FT |
|---|---|---|---|---|---|
| Temporal | $0.262 \pm 0.012$ | $0.273 \pm 0.028$ | $0.249 \pm 0.009$ | $0.291 \pm 0.026$ | $0.317 \pm 0.041$ |
| Random | $0.270 \pm 0.007$ | $0.270 \pm 0.006$ | $0.266 \pm 0.007$ | $0.258 \pm 0.007$ | $\mathbf{0.255 \pm 0.007}$ |

Overall, these results **support overfitting as an explanation to perfomance degradation**. The reasons for overfitting can be both – temporal shift and complex features, as we can see in our benchmark (Sberbank suffers more from temporal shift, KDD does not have shift, but still has performance degradation – which may stem from large feature-to-sample ratio leading to overfitting).

# F DATASETS AND EXTENDED RESULTS

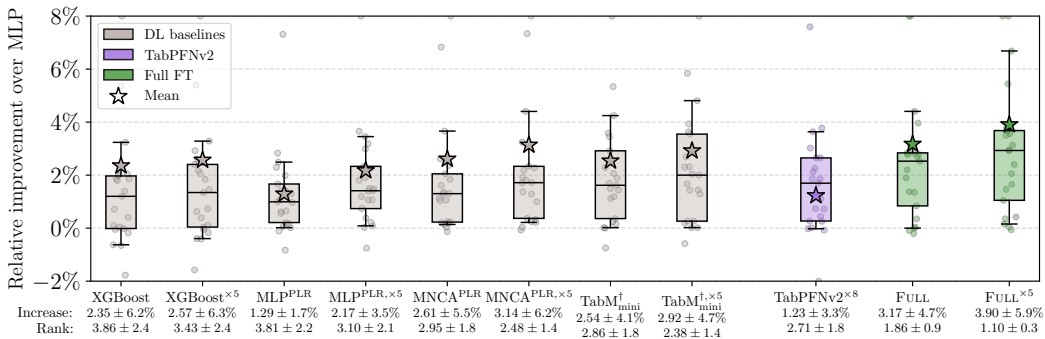

Figure 7: Extended version of Figure 5 with ensembles for all the methods included.

Table 6: The datasets used in our experiments. # num. refers to the number of numerical features, # bin. refers to the number of binary features, # cat. refers to the number of categorical features.

| Name | Train size | Val. size | Test size | # num. | # bin. | # cat. | Task type |
|---|---|---|---|---|---|---|---|
| **Datasets from Grinsztajn et al. (2022)** | | | | | | | |
| wine | 1,787 | 230 | 537 | 11 | 0 | 0 | binary classification |
| phoneme | 2,220 | 285 | 667 | 5 | 0 | 0 | binary classification |
| KDDCup09_upselling | 3,589 | 461 | 1,078 | 34 | 1 | 14 | binary classification |
| kdd_ipums_la_97-small | 3,631 | 467 | 1,090 | 20 | 0 | 0 | binary classification |
| bank-marketing | 7,404 | 952 | 2,222 | 7 | 0 | 0 | binary classification |
| MagicTelescope | 9,363 | 1,203 | 2,810 | 10 | 0 | 0 | binary classification |
| credit | 10,000 | 2,014 | 4,700 | 10 | 0 | 0 | binary classification |
| pol | 10,000 | 1,500 | 3,500 | 26 | 0 | 0 | regression |
| wine_quality | 4,547 | 585 | 1,365 | 11 | 0 | 0 | regression |
| Brazilian_houses | 7,484 | 962 | 2,246 | 8 | 0 | 0 | regression |
| Ailerons | 9,625 | 1,237 | 2,888 | 33 | 0 | 0 | regression |
| MiamiHousing2016 | 9,752 | 1,254 | 2,926 | 13 | 0 | 0 | regression |
| elevators | 10,000 | 1,979 | 4,620 | 16 | 0 | 0 | regression |
| fifa | 10,000 | 2,418 | 5,645 | 5 | 0 | 0 | regression |
| house_sales | 10,000 | 3,483 | 8,130 | 15 | 0 | 0 | regression |
| medical_charges | 10,000 | 45,919 | 50,000 | 3 | 0 | 0 | regression |
| **Datasets from Gorishniy et al. (2021)** | | | | | | | |
| churn | 6,400 | 1,600 | 2,000 | 7 | 3 | 1 | binary classification |
| adult | 26,048 | 6,513 | 16,281 | 6 | 1 | 7 | binary classification |
| california | 13,209 | 3,303 | 4,128 | 8 | 0 | 0 | regression |
| house | 14,581 | 3,646 | 4,557 | 16 | 0 | 0 | regression |
| diamond | 34,521 | 8,631 | 10,788 | 6 | 0 | 3 | regression |
| **Subsamples from TabReD Rubachev et al. (2025)** | | | | | | | |
| Ecom Offers | 8403 | 10000 | 10000 | 113 | 6 | 0 | binary classification |
| Homesite Insurance | 3344 | 10000 | 10000 | 253 | 23 | 23 | binary classifiation |
| Homecredit Default | 1436 | 10000 | 10000 | 612 | 2 | 82 | binary classification |
| Maps Routing | 1014 | 10000 | 10000 | 984 | 0 | 2 | regression |
| Cooking Time | 5208 | 10000 | 10000 | 186 | 3 | 3 | regression |
| Delivery ETA | 4484 | 10000 | 10000 | 221 | 1 | 1 | regression |
| Weather | 9708 | 10000 | 10000 | 100 | 3 | 0 | regression |
| Sberbank Housing | 2551 | 4827 | 4647 | 365 | 17 | 10 | regression |

Table 7: Extended results for the benchmark. Results are grouped by datasets.

| | Ailerons ↓ | | | Brazilian_houses ↓ | |
|---|---|---|---|---|---|
| Method | Single model | Ensemble | Method | Single model | Ensemble |
| MLP | $0.0002 \pm 0.0000$ | $0.0002 \pm 0.0000$ | MLP | $0.0469 \pm 0.0178$ | $0.0440 \pm 0.0207$ |
| XGBoost | $0.0002 \pm 0.0000$ | $0.0002 \pm 0.0000$ | XGBoost | $0.0541 \pm 0.0279$ | $0.0535 \pm 0.0287$ |
| MLP$^{\text{PLR}}$ | $0.0002 \pm 0.0000$ | $0.0002 \pm 0.0000$ | MLP$^{\text{PLR}}$ | $0.0422 \pm 0.0182$ | $0.0397 \pm 0.0206$ |
| MNCA | $0.0002 \pm 0.0000$ | $0.0002 \pm 0.0000$ | MNCA | $0.0525 \pm 0.0160$ | $0.0509 \pm 0.0180$ |
| TabM$^{\dagger}_{\text{mini}}$ | $0.0002 \pm 0.0000$ | $0.0002 \pm 0.0000$ | TabM$^{\dagger}_{\text{mini}}$ | $0.0459 \pm 0.0204$ | $0.0439 \pm 0.0228$ |
| No FT | $0.0002 \pm 0.0000$ | – | No FT | $0.0457 \pm 0.0032$ | – |
| TabPFNv2$^{\times 8}$ | $0.0002 \pm 0.0000$ | – | TabPFNv2$^{\times 8}$ | $0.0199 \pm 0.0210$ | – |
| Last Layers | $0.0001 \pm 0.0000$ | – | Last Layers | $0.0438 \pm 0.0039$ | – |
| LoRA | $0.0002 \pm 0.0000$ | – | LoRA | $0.0447 \pm 0.0038$ | – |
| Full$^{\text{untied}}$ | $0.0001 \pm 0.0000$ | – | Full$^{\text{untied}}$ | $0.0495 \pm 0.0127$ | – |
| Full$^{\text{PLE}}$ | $0.0001 \pm 0.0000$ | – | Full$^{\text{PLE}}$ | $0.0624 \pm 0.0244$ | – |
| Emb.,LN,Head | $0.0001 \pm 0.0000$ | $0.0001 \pm 0.0000$ | Emb.,LN,Head | $0.0465 \pm 0.0026$ | $0.0465 \pm 0.0030$ |
| Full | $0.0001 \pm 0.0000$ | $0.0001 \pm 0.0000$ | Full | $0.0569 \pm 0.0241$ | $0.0506 \pm 0.0228$ |

KDDCup09_upselling ↑

| Method | Single model | Ensemble |
| --- | --- | --- |
| MLP | $0.7763 \pm 0.0150$ | $0.7806 \pm 0.0125$ |
| XGBoost | $0.7922 \pm 0.0114$ | $0.7950 \pm 0.0102$ |
| MLP$^{\text{PLR}}$ | $0.7983 \pm 0.0088$ | $0.7995 \pm 0.0105$ |
| MNCA | $0.7929 \pm 0.0087$ | $0.7989 \pm 0.0115$ |
| TabM$^{\dagger}_{\text{mini}}$ | $0.8042 \pm 0.0144$ | $0.8039 \pm 0.0114$ |
| No FT | $0.8109 \pm 0.0096$ | – |
| TabPFNv2$^{\times 8}$ | $0.8046 \pm 0.0129$ | – |
| Last Layers | $0.8022 \pm 0.0106$ | – |
| LoRA | $0.8077 \pm 0.0053$ | – |
| Full$^{\text{untied}}$ | $0.7995 \pm 0.0124$ | – |
| Full$^{\text{PLE}}$ | $0.7999 \pm 0.0138$ | – |
| Emb.,LN,Head | $0.8054 \pm 0.0062$ | $0.8066 \pm 0.0091$ |
| Full | $0.7983 \pm 0.0087$ | $0.8056 \pm 0.0116$ |

MagicTelescope ↑

| Method | Single model | Ensemble |
| --- | --- | --- |
| MLP | $0.8536 \pm 0.0063$ | $0.8566 \pm 0.0061$ |
| XGBoost | $0.8539 \pm 0.0100$ | $0.8589 \pm 0.0110$ |
| MLP$^{\text{PLR}}$ | $0.8583 \pm 0.0058$ | $0.8626 \pm 0.0044$ |
| MNCA | $0.8580 \pm 0.0059$ | $0.8628 \pm 0.0041$ |
| TabM$^{\dagger}_{\text{mini}}$ | $0.8637 \pm 0.0094$ | $0.8646 \pm 0.0075$ |
| No FT | $0.8647 \pm 0.0059$ | – |
| TabPFNv2$^{\times 8}$ | $0.8695 \pm 0.0073$ | – |
| Last Layers | $0.8765 \pm 0.0051$ | – |
| LoRA | $0.8738 \pm 0.0059$ | – |
| Full$^{\text{untied}}$ | $0.8780 \pm 0.0052$ | – |
| Full$^{\text{PLE}}$ | $0.8778 \pm 0.0073$ | – |
| Emb.,LN,Head | $0.8765 \pm 0.0051$ | $0.8772 \pm 0.0062$ |
| Full | $0.8765 \pm 0.0056$ | $0.8803 \pm 0.0061$ |

MiamiHousing2016 ↓

| Method | Single model | Ensemble |
| --- | --- | --- |
| MLP | $0.1613 \pm 0.0029$ | $0.1574 \pm 0.0043$ |
| XGBoost | $0.1439 \pm 0.0030$ | $0.1434 \pm 0.0029$ |
| MLP$^{\text{PLR}}$ | $0.1519 \pm 0.0028$ | $0.1479 \pm 0.0017$ |
| MNCA | $0.1501 \pm 0.0037$ | $0.1477 \pm 0.0032$ |
| TabM$^{\dagger}_{\text{mini}}$ | $0.1412 \pm 0.0017$ | $0.1387 \pm 0.0008$ |
| No FT | $0.1369 \pm 0.0023$ | – |
| TabPFNv2$^{\times 8}$ | $0.1349 \pm 0.0026$ | – |
| Last Layers | $0.1329 \pm 0.0019$ | – |
| LoRA | $0.1333 \pm 0.0031$ | – |
| Full$^{\text{untied}}$ | $0.1341 \pm 0.0021$ | – |
| Full$^{\text{PLE}}$ | $0.1337 \pm 0.0026$ | – |
| Emb.,LN,Head | $0.1339 \pm 0.0023$ | $0.1334 \pm 0.0028$ |
| Full | $0.1334 \pm 0.0035$ | $0.1316 \pm 0.0031$ |

adult ↑

| Method | Single model | Ensemble |
| --- | --- | --- |
| MLP | $0.8548 \pm 0.0006$ | $0.8559 \pm 0.0011$ |
| XGBoost | $0.8719 \pm 0.0008$ | $0.8723 \pm 0.0002$ |
| MLP$^{\text{PLR}}$ | $0.8690 \pm 0.0006$ | $0.8702 \pm 0.0006$ |
| MNCA | $0.8676 \pm 0.0021$ | $0.8696 \pm 0.0003$ |
| TabM$^{\dagger}_{\text{mini}}$ | $0.8675 \pm 0.0018$ | $0.8690 \pm 0.0005$ |
| No FT | $0.8588 \pm 0.0004$ | – |
| TabPFNv2$^{\times 8}$ | $0.8611 \pm 0.0007$ | – |
| Last Layers | $0.8702 \pm 0.0006$ | – |
| LoRA | $0.8704 \pm 0.0011$ | – |
| Full$^{\text{untied}}$ | $0.8719 \pm 0.0010$ | – |
| Full$^{\text{PLE}}$ | $0.8723 \pm 0.0004$ | – |
| Emb.,LN,Head | $0.8705 \pm 0.0009$ | $0.8713 \pm nan$ |
| Full | $0.8710 \pm 0.0014$ | $0.8723 \pm nan$ |

bank-marketing ↑

| Method | Single model | Ensemble |
| --- | --- | --- |
| MLP | $0.7860 \pm 0.0055$ | $0.7887 \pm 0.0052$ |
| XGBoost | $0.8014 \pm 0.0088$ | $0.8030 \pm 0.0076$ |
| MLP$^{\text{PLR}}$ | $0.7946 \pm 0.0100$ | $0.7977 \pm 0.0117$ |
| MNCA | $0.7955 \pm 0.0075$ | $0.8003 \pm 0.0077$ |
| TabM$^{\dagger}_{\text{mini}}$ | $0.7992 \pm 0.0093$ | $0.8017 \pm 0.0087$ |
| No FT | $0.8025 \pm 0.0078$ | – |
| TabPFNv2$^{\times 8}$ | $0.8026 \pm 0.0075$ | – |
| Last Layers | $0.8031 \pm 0.0099$ | – |
| LoRA | $0.8053 \pm 0.0069$ | – |
| Full$^{\text{untied}}$ | $0.8044 \pm 0.0087$ | – |
| Full$^{\text{PLE}}$ | $0.8013 \pm 0.0077$ | – |
| Emb.,LN,Head | $0.8044 \pm 0.0076$ | $0.8051 \pm 0.0077$ |
| Full | $0.8032 \pm 0.0074$ | $0.8048 \pm 0.0075$ |

california ↓

| Method | Single model | Ensemble |
| --- | --- | --- |
| MLP | $0.4935 \pm 0.0042$ | $0.4880 \pm 0.0022$ |
| XGBoost | $0.4319 \pm 0.0018$ | $0.4316 \pm 0.0007$ |
| MLP$^{\text{PLR}}$ | $0.4659 \pm 0.0035$ | $0.4549 \pm 0.0006$ |
| MNCA | $0.4236 \pm 0.0008$ | $0.4231 \pm 0.0005$ |
| TabM$^{\dagger}_{\text{mini}}$ | $0.4323 \pm 0.0046$ | $0.4261 \pm 0.0019$ |
| No FT | $0.3987 \pm 0.0003$ | – |
| TabPFNv2$^{\times 8}$ | $0.4038 \pm 0.0016$ | – |
| Last Layers | $0.3897 \pm 0.0027$ | – |
| LoRA | $0.3836 \pm 0.0010$ | – |
| Full$^{\text{untied}}$ | $0.3827 \pm 0.0038$ | – |
| Full$^{\text{PLE}}$ | $0.3843 \pm 0.0024$ | – |
| Emb.,LN,Head | $0.3880 \pm 0.0014$ | $0.3862 \pm nan$ |
| Full | $0.3822 \pm 0.0011$ | $0.3789 \pm nan$ |

| churn ↑ | | |
| --- | --- | --- |
| Method | Single model | Ensemble |
| MLP | $0.8575 \pm 0.0028$ | $0.8582 \pm 0.0008$ |
| XGBoost | $0.8610 \pm 0.0018$ | $0.8608 \pm 0.0013$ |
| MLP$^{PLR}$ | $0.8628 \pm 0.0009$ | $0.8638 \pm 0.0012$ |
| MNCA | $0.8584 \pm 0.0023$ | $0.8615 \pm 0.0013$ |
| TabM$^\dagger_{mini}$ | $0.8606 \pm 0.0023$ | $0.8592 \pm 0.0003$ |
| No FT | $0.8590 \pm 0.0008$ | – |
| TabPFNv2$^{\times 8}$ | $0.8637 \pm 0.0012$ | – |
| Last Layers | $0.8628 \pm 0.0021$ | – |
| LoRA | $0.8609 \pm 0.0018$ | – |
| Full$^{untied}$ | $0.8631 \pm 0.0026$ | – |
| Full$^{PLE}$ | $0.8631 \pm 0.0015$ | – |
| Emb.,LN,Head | $0.8605 \pm 0.0023$ | $0.8605 \pm nan$ |
| Full | $0.8647 \pm 0.0028$ | $0.8665 \pm nan$ |

| credit ↑ | | |
| --- | --- | --- |
| Method | Single model | Ensemble |
| MLP | $0.7737 \pm 0.0052$ | $0.7729 \pm 0.0047$ |
| XGBoost | $0.7688 \pm 0.0025$ | $0.7706 \pm 0.0029$ |
| MLP$^{PLR}$ | $0.7753 \pm 0.0053$ | $0.7767 \pm 0.0075$ |
| MNCA | $0.7737 \pm 0.0033$ | $0.7757 \pm 0.0026$ |
| TabM$^\dagger_{mini}$ | $0.7749 \pm 0.0031$ | $0.7757 \pm 0.0036$ |
| No FT | $0.7746 \pm 0.0019$ | – |
| TabPFNv2$^{\times 8}$ | $0.7735 \pm 0.0030$ | – |
| Last Layers | $0.7720 \pm 0.0044$ | – |
| LoRA | $0.7746 \pm 0.0032$ | – |
| Full$^{untied}$ | $0.7743 \pm 0.0039$ | – |
| Full$^{PLE}$ | $0.7737 \pm 0.0041$ | – |
| Emb.,LN,Head | $0.7756 \pm 0.0036$ | $0.7754 \pm 0.0041$ |
| Full | $0.7730 \pm 0.0035$ | $0.7749 \pm 0.0030$ |

| diamond ↓ | | |
| --- | --- | --- |
| Method | Single model | Ensemble |
| MLP | $0.1402 \pm 0.0016$ | $0.1362 \pm 0.0003$ |
| XGBoost | $0.1368 \pm 0.0002$ | $0.1363 \pm 0.0001$ |
| MLP$^{PLR}$ | $0.1341 \pm 0.0009$ | $0.1325 \pm 0.0004$ |
| MNCA | $0.1368 \pm 0.0010$ | $0.1348 \pm 0.0005$ |
| TabM$^\dagger_{mini}$ | $0.1314 \pm 0.0011$ | $0.1307 \pm 0.0005$ |
| No FT | $0.1370 \pm 0.0002$ | – |
| TabPFNv2$^{\times 8}$ | $0.1311 \pm 0.0005$ | – |
| Last Layers | $0.1302 \pm 0.0006$ | – |
| LoRA | $0.1300 \pm 0.0017$ | – |
| Full$^{untied}$ | $0.1285 \pm 0.0009$ | – |
| Full$^{PLE}$ | $0.1275 \pm 0.0011$ | – |
| Emb.,LN,Head | $0.1323 \pm 0.0006$ | $0.1320 \pm nan$ |
| Full | $0.1275 \pm 0.0007$ | $0.1245 \pm nan$ |

| elevators ↓ | | |
| --- | --- | --- |
| Method | Single model | Ensemble |
| MLP | $0.0020 \pm 0.0000$ | $0.0019 \pm 0.0000$ |
| XGBoost | $0.0020 \pm 0.0000$ | $0.0020 \pm 0.0000$ |
| MLP$^{PLR}$ | $0.0018 \pm 0.0000$ | $0.0018 \pm 0.0000$ |
| MNCA | $0.0019 \pm 0.0000$ | $0.0019 \pm 0.0000$ |
| TabM$^\dagger_{mini}$ | $0.0018 \pm 0.0000$ | $0.0018 \pm 0.0000$ |
| No FT | $0.0019 \pm 0.0000$ | – |
| TabPFNv2$^{\times 8}$ | $0.0019 \pm 0.0000$ | – |
| Last Layers | $0.0018 \pm 0.0000$ | – |
| LoRA | $0.0018 \pm 0.0000$ | – |
| Full$^{untied}$ | $0.0018 \pm 0.0000$ | – |
| Full$^{PLE}$ | $0.0018 \pm 0.0000$ | – |
| Emb.,LN,Head | $0.0018 \pm 0.0000$ | $0.0018 \pm 0.0000$ |
| Full | $0.0018 \pm 0.0000$ | $0.0018 \pm 0.0000$ |

| fifa ↓ | | |
| --- | --- | --- |
| Method | Single model | Ensemble |
| MLP | $0.8038 \pm 0.0125$ | $0.8011 \pm 0.0143$ |
| XGBoost | $0.7799 \pm 0.0110$ | $0.7795 \pm 0.0114$ |
| MLP$^{PLR}$ | $0.7935 \pm 0.0127$ | $0.7898 \pm 0.0141$ |
| MNCA | $0.7956 \pm 0.0140$ | $0.7933 \pm 0.0145$ |
| TabM$^\dagger_{mini}$ | $0.7783 \pm 0.0128$ | $0.7768 \pm 0.0123$ |
| No FT | $0.7833 \pm 0.0085$ | – |
| TabPFNv2$^{\times 8}$ | $0.7815 \pm 0.0106$ | – |
| Last Layers | $0.7820 \pm 0.0150$ | – |
| LoRA | $0.7834 \pm 0.0085$ | – |
| Full$^{untied}$ | $0.7845 \pm 0.0126$ | – |
| Full$^{PLE}$ | $0.7818 \pm 0.0111$ | – |
| Emb.,LN,Head | $0.7773 \pm 0.0126$ | $0.7764 \pm 0.0168$ |
| Full | $0.7834 \pm 0.0106$ | $0.7779 \pm 0.0127$ |

| house ↓ | | |
| --- | --- | --- |
| Method | Single model | Ensemble |
| MLP | $3.1163 \pm 0.0248$ | $3.0706 \pm 0.0140$ |
| XGBoost | $3.1703 \pm 0.0098$ | $3.1644 \pm 0.0068$ |
| MLP$^{PLR}$ | $3.0546 \pm 0.0288$ | $3.0170 \pm 0.0070$ |
| MNCA | $3.0928 \pm 0.0340$ | $3.0538 \pm 0.0072$ |
| TabM$^\dagger_{mini}$ | $2.9829 \pm 0.0225$ | $2.9648 \pm 0.0035$ |
| No FT | $3.1100 \pm 0.0053$ | – |
| TabPFNv2$^{\times 8}$ | $3.0637 \pm 0.0045$ | – |
| Last Layers | $2.9826 \pm 0.0335$ | – |
| LoRA | $2.9901 \pm 0.0235$ | – |
| Full$^{untied}$ | $2.9696 \pm 0.0275$ | – |
| Full$^{PLE}$ | $2.9783 \pm 0.0491$ | – |
| Emb.,LN,Head | $3.0748 \pm 0.0135$ | $3.0660 \pm nan$ |
| Full | $2.9919 \pm 0.0268$ | $2.9036 \pm nan$ |

| house_sales ↓ | | |
|---|---|---|
| Method | Single model | Ensemble |
| MLP | $0.1791 \pm 0.0009$ | $0.1763 \pm 0.0003$ |
| XGBoost | $0.1693 \pm 0.0002$ | $0.1689 \pm 0.0001$ |
| MLP$^{\text{PLR}}$ | $0.1693 \pm 0.0005$ | $0.1687 \pm 0.0007$ |
| MNCA | $0.1740 \pm 0.0018$ | $0.1714 \pm 0.0005$ |
| TabM$^{\dagger}_{\text{mini}}$ | $0.1658 \pm 0.0005$ | $0.1647 \pm 0.0002$ |
| No FT | $0.1632 \pm 0.0000$ | – |
| TabPFNv2$^{\times 8}$ | $0.1601 \pm 0.0001$ | – |
| Last Layers | $0.1612 \pm 0.0003$ | – |
| LoRA | $0.1592 \pm 0.0002$ | – |
| Full$^{\text{untied}}$ | $0.1589 \pm 0.0006$ | – |
| Full$^{\text{PLE}}$ | $0.1588 \pm 0.0004$ | – |
| Emb.,LN,Head | $0.1605 \pm 0.0001$ | $0.1603 \pm nan$ |
| Full | $0.1586 \pm 0.0005$ | $0.1579 \pm nan$ |

| kdd_ipums_la_97-small ↑ | | |
|---|---|---|
| Method | Single model | Ensemble |
| MLP | $0.8831 \pm 0.0068$ | $0.8845 \pm 0.0055$ |
| XGBoost | $0.8830 \pm 0.0086$ | $0.8835 \pm 0.0085$ |
| MLP$^{\text{PLR}}$ | $0.8758 \pm 0.0112$ | $0.8765 \pm 0.0108$ |
| MNCA | $0.8807 \pm 0.0046$ | $0.8832 \pm 0.0048$ |
| TabM$^{\dagger}_{\text{mini}}$ | $0.8765 \pm 0.0091$ | $0.8780 \pm 0.0099$ |
| No FT | $0.8810 \pm 0.0028$ | – |
| TabPFNv2$^{\times 8}$ | $0.8824 \pm 0.0059$ | – |
| Last Layers | $0.8828 \pm 0.0054$ | – |
| LoRA | $0.8823 \pm 0.0045$ | – |
| Full$^{\text{untied}}$ | $0.8761 \pm 0.0153$ | – |
| Full$^{\text{PLE}}$ | $0.8840 \pm 0.0061$ | – |
| Emb.,LN,Head | $0.8821 \pm 0.0041$ | $0.8820 \pm 0.0055$ |
| Full | $0.8831 \pm 0.0063$ | $0.8862 \pm 0.0078$ |

| medical_charges ↓ | | |
|---|---|---|
| Method | Single model | Ensemble |
| MLP | $0.0816 \pm 0.0002$ | $0.0814 \pm 0.0000$ |
| XGBoost | $0.0825 \pm 0.0001$ | $0.0825 \pm 0.0000$ |
| MLP$^{\text{PLR}}$ | $0.0812 \pm 0.0002$ | $0.0810 \pm 0.0000$ |
| MNCA | $0.0811 \pm 0.0000$ | $0.0810 \pm 0.0000$ |
| TabM$^{\dagger}_{\text{mini}}$ | $0.0812 \pm 0.0001$ | $0.0812 \pm 0.0000$ |
| No FT | $0.0812 \pm 0.0000$ | – |
| TabPFNv2$^{\times 8}$ | $0.0813 \pm 0.0000$ | – |
| Last Layers | $0.0808 \pm 0.0000$ | – |
| LoRA | $0.0809 \pm 0.0000$ | – |
| Full$^{\text{untied}}$ | $0.0809 \pm 0.0000$ | – |
| Full$^{\text{PLE}}$ | $0.0808 \pm 0.0000$ | – |
| Emb.,LN,Head | $0.0809 \pm 0.0000$ | $0.0808 \pm nan$ |
| Full | $0.0808 \pm 0.0000$ | $0.0807 \pm nan$ |

| phoneme ↑ | | |
|---|---|---|
| Method | Single model | Ensemble |
| MLP | $0.8548 \pm 0.0132$ | $0.8635 \pm 0.0099$ |
| XGBoost | $0.8708 \pm 0.0134$ | $0.8771 \pm 0.0156$ |
| MLP$^{\text{PLR}}$ | $0.8744 \pm 0.0105$ | $0.8861 \pm 0.0071$ |
| MNCA | $0.8810 \pm 0.0090$ | $0.8861 \pm 0.0057$ |
| TabM$^{\dagger}_{\text{mini}}$ | $0.8798 \pm 0.0088$ | $0.8885 \pm 0.0056$ |
| No FT | $0.8837 \pm 0.0074$ | – |
| TabPFNv2$^{\times 8}$ | $0.8871 \pm 0.0064$ | – |
| Last Layers | $0.8883 \pm 0.0098$ | – |
| LoRA | $0.8919 \pm 0.0073$ | – |
| Full$^{\text{untied}}$ | $0.8721 \pm 0.0566$ | – |
| Full$^{\text{PLE}}$ | $0.8916 \pm 0.0085$ | – |
| Emb.,LN,Head | $0.8913 \pm 0.0082$ | $0.8918 \pm 0.0105$ |
| Full | $0.8925 \pm 0.0104$ | $0.9013 \pm 0.0071$ |

| pol ↓ | | |
|---|---|---|
| Method | Single model | Ensemble |
| MLP | $5.5216 \pm 0.6947$ | $4.9945 \pm 0.5923$ |
| XGBoost | $4.3030 \pm 0.0677$ | $4.2548 \pm 0.0488$ |
| MLP$^{\text{PLR}}$ | $2.8846 \pm 0.3192$ | $2.5266 \pm 0.0605$ |
| MNCA | $5.7569 \pm 0.5465$ | $5.3773 \pm 0.5463$ |
| TabM$^{\dagger}_{\text{mini}}$ | $2.4521 \pm 0.1371$ | $2.4175 \pm 0.1124$ |
| No FT | $4.8233 \pm 0.0533$ | – |
| TabPFNv2$^{\times 8}$ | $3.4119 \pm 0.1679$ | – |
| Last Layers | $2.8283 \pm 0.2159$ | – |
| LoRA | $2.5989 \pm 0.0357$ | – |
| Full$^{\text{untied}}$ | $5.3104 \pm 7.8499$ | – |
| Full$^{\text{PLE}}$ | $2.7128 \pm 0.1078$ | – |
| Emb.,LN,Head | $2.8319 \pm 0.2250$ | $2.7829 \pm 0.2584$ |
| Full | $2.6424 \pm 0.1172$ | $2.3375 \pm 0.1385$ |

| wine ↑ | | |
|---|---|---|
| Method | Single model | Ensemble |
| MLP | $0.7782 \pm 0.0145$ | $0.7907 \pm 0.0117$ |
| XGBoost | $0.7927 \pm 0.0209$ | $0.8010 \pm 0.0186$ |
| MLP$^{\text{PLR}}$ | $0.7774 \pm 0.0154$ | $0.7964 \pm 0.0146$ |
| MNCA | $0.7879 \pm 0.0150$ | $0.8005 \pm 0.0121$ |
| TabM$^{\dagger}_{\text{mini}}$ | $0.7908 \pm 0.0167$ | $0.7963 \pm 0.0113$ |
| No FT | $0.7865 \pm 0.0132$ | – |
| TabPFNv2$^{\times 8}$ | $0.7958 \pm 0.0092$ | – |
| Last Layers | $0.7983 \pm 0.0195$ | – |
| LoRA | $0.8024 \pm 0.0137$ | – |
| Full$^{\text{untied}}$ | $0.8024 \pm 0.0152$ | – |
| Full$^{\text{PLE}}$ | $0.8020 \pm 0.0141$ | – |
| Emb.,LN,Head | $0.8021 \pm 0.0131$ | $0.8074 \pm 0.0145$ |
| Full | $0.7979 \pm 0.0139$ | $0.8060 \pm 0.0118$ |

| | wine_quality $\downarrow$ | |
| Method | Single model | Ensemble |
| --- | --- | --- |
| MLP | $0.6703 \pm 0.0170$ | $0.6530 \pm 0.0152$ |
| XGBoost | $0.6035 \pm 0.0142$ | $0.6025 \pm 0.0139$ |
| MLP$^{\mathrm{PLR}}$ | $0.6537 \pm 0.0235$ | $0.6328 \pm 0.0155$ |
| MNCA | $0.6151 \pm 0.0092$ | $0.6058 \pm 0.0149$ |
| TabM$^{\dagger}_{\mathrm{mini}}$ | $0.6270 \pm 0.0153$ | $0.6194 \pm 0.0150$ |
| No FT | $0.6841 \pm 0.0209$ | – |
| TabPFNv2$^{\times 8}$ | $0.6932 \pm 0.0244$ | – |
| Last Layers | $0.6245 \pm 0.0073$ | – |
| LoRA | $0.6160 \pm 0.0108$ | – |
| Full$^{\mathrm{untied}}$ | $0.6224 \pm 0.0117$ | – |
| Full$^{\mathrm{PLE}}$ | $0.6216 \pm 0.0109$ | – |
| Emb.,LN,Head | $0.6115 \pm 0.0174$ | $0.6077 \pm 0.0188$ |
| Full | $0.6206 \pm 0.0099$ | $0.6068 \pm 0.0145$ |

Table 8: Extended results for the subsampled TabReD benchmark. Results are grouped by datasets.

| sberbank-housing ↓ | | | ecom-offers ↑ | | |
|---|---|---|---|---|---|
| Method | Single model | Ensemble | Method | Single model | Ensemble |
| **split-0** | | | **split-0** | | |
| MLP | $0.2530 \pm 0.0075$ | – | MLP | $0.5986 \pm 0.0032$ | – |
| MLP$^{\text{PLR}}$ | $0.2473 \pm 0.0010$ | – | MLP$^{\text{PLR}}$ | $0.5955 \pm 0.0060$ | – |
| MNCA | $0.2512 \pm 0.0017$ | – | MNCA | $0.5876 \pm 0.0011$ | – |
| TabM$^{\dagger}_{\text{mini}}$ | $0.2393 \pm 0.0032$ | – | TabM$^{\dagger}_{\text{mini}}$ | $0.5895 \pm 0.0042$ | – |
| No FT | $0.3020 \pm 0.0026$ | – | No FT | $0.5595 \pm 0.0006$ | – |
| Full | $0.3552 \pm 0.0213$ | – | Full | $0.5808 \pm 0.0099$ | – |
| **split-1** | | | **split-1** | | |
| MLP | $0.2650 \pm 0.0113$ | – | MLP | $0.6031 \pm 0.0012$ | – |
| MLP$^{\text{PLR}}$ | $0.2593 \pm 0.0068$ | – | MLP$^{\text{PLR}}$ | $0.5913 \pm 0.0070$ | – |
| MNCA | $0.2907 \pm 0.0128$ | – | MNCA | $0.5961 \pm 0.0074$ | – |
| TabM$^{\dagger}_{\text{mini}}$ | $0.2611 \pm 0.0057$ | – | TabM$^{\dagger}_{\text{mini}}$ | $0.6024 \pm 0.0030$ | – |
| No FT | $0.2980 \pm 0.0028$ | – | No FT | $0.5514 \pm 0.0006$ | – |
| Full | $0.3090 \pm 0.0271$ | – | Full | $0.5930 \pm 0.0054$ | – |
| **split-2** | | | **split-2** | | |
| MLP | $0.2621 \pm 0.0060$ | – | MLP | $0.6021 \pm 0.0026$ | – |
| MLP$^{\text{PLR}}$ | $0.2573 \pm 0.0026$ | – | MLP$^{\text{PLR}}$ | $0.5939 \pm 0.0094$ | – |
| MNCA | $0.2821 \pm 0.0239$ | – | MNCA | $0.5858 \pm 0.0058$ | – |
| TabM$^{\dagger}_{\text{mini}}$ | $0.2553 \pm 0.0064$ | – | TabM$^{\dagger}_{\text{mini}}$ | $0.5944 \pm 0.0023$ | – |
| No FT | $0.2808 \pm 0.0020$ | – | No FT | $0.5570 \pm 0.0008$ | – |
| Full | $0.3113 \pm 0.0309$ | – | Full | $0.5767 \pm 0.0359$ | – |
| **split-3** | | | **split-3** | | |
| MLP | $0.2656 \pm 0.0124$ | – | MLP | $0.6076 \pm 0.0012$ | – |
| MLP$^{\text{PLR}}$ | $0.2461 \pm 0.0007$ | – | MLP$^{\text{PLR}}$ | $0.5921 \pm 0.0127$ | – |
| MNCA | $0.2838 \pm 0.0495$ | – | MNCA | $0.5924 \pm 0.0145$ | – |
| TabM$^{\dagger}_{\text{mini}}$ | $0.2454 \pm 0.0034$ | – | TabM$^{\dagger}_{\text{mini}}$ | $0.5952 \pm 0.0022$ | – |
| No FT | $0.3237 \pm 0.0026$ | – | No FT | $0.5608 \pm 0.0010$ | – |
| Full | $0.3473 \pm 0.0269$ | – | Full | $0.5958 \pm 0.0062$ | – |
| **split-4** | | | **split-4** | | |
| MLP | $0.2624 \pm 0.0177$ | – | MLP | $0.6010 \pm 0.0011$ | – |
| MLP$^{\text{PLR}}$ | $0.2482 \pm 0.0017$ | – | MLP$^{\text{PLR}}$ | $0.5939 \pm 0.0015$ | – |
| MNCA | $0.2580 \pm 0.0063$ | – | MNCA | $0.5830 \pm 0.0086$ | – |
| TabM$^{\dagger}_{\text{mini}}$ | $0.2433 \pm 0.0018$ | – | TabM$^{\dagger}_{\text{mini}}$ | $0.5890 \pm 0.0039$ | – |
| No FT | $0.2484 \pm 0.0005$ | – | No FT | $0.5538 \pm 0.0007$ | – |
| Full | $0.2608 \pm 0.0083$ | – | Full | $0.5741 \pm 0.0086$ | – |

| maps-routing ↓ | | | homesite-insurance ↑ | | |
|---|---|---|---|---|---|
| Method | Single model | Ensemble | Method | Single model | Ensemble |
| split-0 | | | split-0 | | |
| MLP | $0.1901 \pm 0.0015$ | – | MLP | $0.9067 \pm 0.0014$ | – |
| $\text{MLP}^{\text{PLR}}$ | $0.1915 \pm 0.0058$ | – | $\text{MLP}^{\text{PLR}}$ | $0.9282 \pm 0.0019$ | – |
| MNCA | $0.2032 \pm 0.0023$ | – | MNCA | $0.9096 \pm 0.0019$ | – |
| $\text{TabM}^{\dagger}_{\text{mini}}$ | $0.1946 \pm 0.0017$ | – | $\text{TabM}^{\dagger}_{\text{mini}}$ | $0.9399 \pm 0.0016$ | – |
| No FT | $0.1717 \pm 0.0000$ | – | No FT | $0.9510 \pm 0.0001$ | – |
| Full | $0.1703 \pm 0.0002$ | – | Full | $0.9538 \pm 0.0011$ | – |
| split-1 | | | split-1 | | |
| MLP | $0.1814 \pm 0.0005$ | – | MLP | $0.9007 \pm 0.0014$ | – |
| $\text{MLP}^{\text{PLR}}$ | $0.1793 \pm 0.0008$ | – | $\text{MLP}^{\text{PLR}}$ | $0.9120 \pm 0.0035$ | – |
| MNCA | $0.1815 \pm 0.0004$ | – | MNCA | $0.8999 \pm 0.0027$ | – |
| $\text{TabM}^{\dagger}_{\text{mini}}$ | $0.1787 \pm 0.0003$ | – | $\text{TabM}^{\dagger}_{\text{mini}}$ | $0.9355 \pm 0.0009$ | – |
| No FT | $0.1776 \pm 0.0000$ | – | No FT | $0.9437 \pm 0.0001$ | – |
| Full | $0.1764 \pm 0.0004$ | – | Full | $0.9480 \pm 0.0027$ | – |
| split-2 | | | split-2 | | |
| MLP | $0.1778 \pm 0.0009$ | – | MLP | $0.9067 \pm 0.0011$ | – |
| $\text{MLP}^{\text{PLR}}$ | $0.1761 \pm 0.0013$ | – | $\text{MLP}^{\text{PLR}}$ | $0.8537 \pm 0.1013$ | – |
| MNCA | $0.1787 \pm 0.0008$ | – | MNCA | $0.9060 \pm 0.0025$ | – |
| $\text{TabM}^{\dagger}_{\text{mini}}$ | $0.1760 \pm 0.0003$ | – | $\text{TabM}^{\dagger}_{\text{mini}}$ | $0.9406 \pm 0.0012$ | – |
| No FT | $0.1718 \pm 0.0000$ | – | No FT | $0.9488 \pm 0.0001$ | – |
| Full | $0.1736 \pm 0.0011$ | – | Full | $0.9521 \pm 0.0007$ | – |
| split-3 | | | split-3 | | |
| MLP | $0.1810 \pm 0.0007$ | – | MLP | $0.9019 \pm 0.0011$ | – |
| $\text{MLP}^{\text{PLR}}$ | $0.1797 \pm 0.0011$ | – | $\text{MLP}^{\text{PLR}}$ | $0.9187 \pm 0.0021$ | – |
| MNCA | $0.1828 \pm 0.0003$ | – | MNCA | $0.9048 \pm 0.0028$ | – |
| $\text{TabM}^{\dagger}_{\text{mini}}$ | $0.1786 \pm 0.0010$ | – | $\text{TabM}^{\dagger}_{\text{mini}}$ | $0.9431 \pm 0.0008$ | – |
| No FT | $0.1766 \pm 0.0001$ | – | No FT | $0.9472 \pm 0.0001$ | – |
| Full | $0.1761 \pm 0.0012$ | – | Full | $0.9519 \pm 0.0014$ | – |
| split-4 | | | split-4 | | |
| MLP | $0.1781 \pm 0.0004$ | – | MLP | $0.9087 \pm 0.0010$ | – |
| $\text{MLP}^{\text{PLR}}$ | $0.1761 \pm 0.0005$ | – | $\text{MLP}^{\text{PLR}}$ | $0.9239 \pm 0.0026$ | – |
| MNCA | $0.1783 \pm 0.0004$ | – | MNCA | $0.9077 \pm 0.0033$ | – |
| $\text{TabM}^{\dagger}_{\text{mini}}$ | $0.1756 \pm 0.0003$ | – | $\text{TabM}^{\dagger}_{\text{mini}}$ | $0.9423 \pm 0.0012$ | – |
| No FT | $0.1729 \pm 0.0000$ | – | No FT | $0.9441 \pm 0.0001$ | – |
| Full | $0.1729 \pm 0.0006$ | – | Full | $0.9489 \pm 0.0013$ | – |

| cooking-time ↓ | | | | homecredit-default ↑ | | |
|---|---|---|---|---|---|---|
| Method | Single model | Ensemble | | Method | Single model | Ensemble |
| **split-0** | | | | **split-0** | | |
| MLP | $0.4893 \pm 0.0005$ | – | | MLP | $0.7788 \pm 0.0037$ | – |
| $\text{MLP}^{\text{PLR}}$ | $0.4819 \pm 0.0011$ | – | | $\text{MLP}^{\text{PLR}}$ | $0.7721 \pm 0.0137$ | – |
| MNCA | $0.4887 \pm 0.0021$ | – | | MNCA | $0.7935 \pm 0.0062$ | – |
| $\text{TabM}^{\dagger}_{\text{mini}}$ | $0.4816 \pm 0.0010$ | – | | $\text{TabM}^{\dagger}_{\text{mini}}$ | $0.7774 \pm 0.0081$ | – |
| No FT | $0.6283 \pm 0.0244$ | – | | No FT | $0.7304 \pm 0.0011$ | – |
| Full | $0.4860 \pm 0.0006$ | – | | Full | $0.7333 \pm 0.0137$ | – |
| **split-1** | | | | **split-1** | | |
| MLP | $0.4950 \pm 0.0012$ | – | | MLP | $0.7407 \pm 0.0021$ | – |
| $\text{MLP}^{\text{PLR}}$ | $0.4926 \pm 0.0011$ | – | | $\text{MLP}^{\text{PLR}}$ | $0.7543 \pm 0.0063$ | – |
| MNCA | $0.5010 \pm 0.0005$ | – | | MNCA | $0.7557 \pm 0.0032$ | – |
| $\text{TabM}^{\dagger}_{\text{mini}}$ | $0.4916 \pm 0.0012$ | – | | $\text{TabM}^{\dagger}_{\text{mini}}$ | $0.7629 \pm 0.0043$ | – |
| No FT | $0.4958 \pm 0.0001$ | – | | No FT | $0.7301 \pm 0.0012$ | – |
| Full | $0.4938 \pm 0.0027$ | – | | Full | $0.7493 \pm 0.0073$ | – |
| **split-2** | | | | **split-2** | | |
| MLP | $0.4927 \pm 0.0007$ | – | | MLP | $0.7739 \pm 0.0047$ | – |
| $\text{MLP}^{\text{PLR}}$ | $0.4910 \pm 0.0007$ | – | | $\text{MLP}^{\text{PLR}}$ | $0.7761 \pm 0.0055$ | – |
| MNCA | $0.4957 \pm 0.0020$ | – | | MNCA | $0.7635 \pm 0.0051$ | – |
| $\text{TabM}^{\dagger}_{\text{mini}}$ | $0.4881 \pm 0.0005$ | – | | $\text{TabM}^{\dagger}_{\text{mini}}$ | $0.7785 \pm 0.0020$ | – |
| No FT | $0.4867 \pm 0.0000$ | – | | No FT | $0.7722 \pm 0.0005$ | – |
| Full | $0.4861 \pm 0.0009$ | – | | Full | $0.7688 \pm 0.0028$ | – |
| **split-3** | | | | **split-3** | | |
| MLP | $0.4967 \pm 0.0007$ | – | | MLP | $0.7569 \pm 0.0035$ | – |
| $\text{MLP}^{\text{PLR}}$ | $0.4917 \pm 0.0009$ | – | | $\text{MLP}^{\text{PLR}}$ | $0.7694 \pm 0.0057$ | – |
| MNCA | $0.4981 \pm 0.0009$ | – | | MNCA | $0.7663 \pm 0.0048$ | – |
| $\text{TabM}^{\dagger}_{\text{mini}}$ | $0.4904 \pm 0.0006$ | – | | $\text{TabM}^{\dagger}_{\text{mini}}$ | $0.7726 \pm 0.0040$ | – |
| No FT | $0.4889 \pm 0.0000$ | – | | No FT | $0.6995 \pm 0.0013$ | – |
| Full | $0.4886 \pm 0.0013$ | – | | Full | $0.7169 \pm 0.0169$ | – |
| **split-4** | | | | **split-4** | | |
| MLP | $0.4972 \pm 0.0009$ | – | | MLP | $0.7500 \pm 0.0023$ | – |
| $\text{MLP}^{\text{PLR}}$ | $0.4932 \pm 0.0016$ | – | | $\text{MLP}^{\text{PLR}}$ | $0.7561 \pm 0.0083$ | – |
| MNCA | $0.5009 \pm 0.0022$ | – | | MNCA | $0.7492 \pm 0.0081$ | – |
| $\text{TabM}^{\dagger}_{\text{mini}}$ | $0.4920 \pm 0.0014$ | – | | $\text{TabM}^{\dagger}_{\text{mini}}$ | $0.7820 \pm 0.0071$ | – |
| No FT | $0.4903 \pm 0.0000$ | – | | No FT | $0.7386 \pm 0.0003$ | – |
| Full | $0.4894 \pm 0.0003$ | – | | Full | $0.7556 \pm 0.0193$ | – |

| delivery-eta ↓ | | | weather ↓ | | |
|---|---|---|---|---|---|
| Method | Single model | Ensemble | Method | Single model | Ensemble |
| | split-0 | | | split-0 | |
| MLP | $0.5742 \pm 0.0053$ | – | MLP | $1.7852 \pm 0.0068$ | – |
| $\text{MLP}^{\text{PLR}}$ | $0.5692 \pm 0.0017$ | – | $\text{MLP}^{\text{PLR}}$ | $1.7235 \pm 0.0192$ | – |
| MNCA | $0.5708 \pm 0.0025$ | – | MNCA | $1.7741 \pm 0.0111$ | – |
| $\text{TabM}^{\dagger}_{\text{mini}}$ | $0.5695 \pm 0.0045$ | – | $\text{TabM}^{\dagger}_{\text{mini}}$ | $1.6804 \pm 0.0049$ | – |
| No FT | $0.5671 \pm 0.0002$ | – | No FT | $1.7334 \pm 0.0004$ | – |
| FULL | $0.5630 \pm 0.0014$ | – | FULL | $1.6129 \pm 0.0095$ | – |
| | split-1 | | | split-1 | |
| MLP | $0.5684 \pm 0.0009$ | – | MLP | $1.6970 \pm 0.0082$ | – |
| $\text{MLP}^{\text{PLR}}$ | $0.5628 \pm 0.0024$ | – | $\text{MLP}^{\text{PLR}}$ | $1.6389 \pm 0.0042$ | – |
| MNCA | $0.5648 \pm 0.0002$ | – | MNCA | $1.6805 \pm 0.0079$ | – |
| $\text{TabM}^{\dagger}_{\text{mini}}$ | $0.5662 \pm 0.0020$ | – | $\text{TabM}^{\dagger}_{\text{mini}}$ | $1.6167 \pm 0.0028$ | – |
| No FT | $0.5629 \pm 0.0001$ | – | No FT | $1.7161 \pm 0.0003$ | – |
| FULL | $0.5642 \pm 0.0014$ | – | FULL | $1.6241 \pm 0.0105$ | – |
| | split-2 | | | split-2 | |
| MLP | $0.5803 \pm 0.0076$ | – | MLP | $1.6961 \pm 0.0098$ | – |
| $\text{MLP}^{\text{PLR}}$ | $0.5636 \pm 0.0020$ | – | $\text{MLP}^{\text{PLR}}$ | $1.6355 \pm 0.0049$ | – |
| MNCA | $0.5651 \pm 0.0026$ | – | MNCA | $1.6787 \pm 0.0032$ | – |
| $\text{TabM}^{\dagger}_{\text{mini}}$ | $0.5600 \pm 0.0028$ | – | $\text{TabM}^{\dagger}_{\text{mini}}$ | $1.6164 \pm 0.0049$ | – |
| No FT | $0.5632 \pm 0.0001$ | – | No FT | $1.7307 \pm 0.0005$ | – |
| FULL | $0.5607 \pm 0.0018$ | – | FULL | $1.6216 \pm 0.0110$ | – |
| | split-3 | | | split-3 | |
| MLP | $0.5587 \pm 0.0013$ | – | MLP | $1.6694 \pm 0.0042$ | – |
| $\text{MLP}^{\text{PLR}}$ | $0.5537 \pm 0.0012$ | – | $\text{MLP}^{\text{PLR}}$ | $1.6098 \pm 0.0028$ | – |
| MNCA | $0.5588 \pm 0.0012$ | – | MNCA | $1.6548 \pm 0.0047$ | – |
| $\text{TabM}^{\dagger}_{\text{mini}}$ | $0.5564 \pm 0.0022$ | – | $\text{TabM}^{\dagger}_{\text{mini}}$ | $1.5806 \pm 0.0061$ | – |
| No FT | $0.5516 \pm 0.0000$ | – | No FT | $1.7011 \pm 0.0010$ | – |
| FULL | $0.5518 \pm 0.0017$ | – | FULL | $1.5921 \pm 0.0111$ | – |
| | split-4 | | | split-4 | |
| MLP | $0.5630 \pm 0.0011$ | – | MLP | $1.7196 \pm 0.0064$ | – |
| $\text{MLP}^{\text{PLR}}$ | $0.5611 \pm 0.0015$ | – | $\text{MLP}^{\text{PLR}}$ | $1.6567 \pm 0.0042$ | – |
| MNCA | $0.5623 \pm 0.0015$ | – | MNCA | $1.6825 \pm 0.0038$ | – |
| $\text{TabM}^{\dagger}_{\text{mini}}$ | $0.5566 \pm 0.0018$ | – | $\text{TabM}^{\dagger}_{\text{mini}}$ | $1.6213 \pm 0.0062$ | – |
| No FT | $0.5649 \pm 0.0002$ | – | No FT | $1.7453 \pm 0.0014$ | – |
| FULL | $0.5588 \pm 0.0011$ | – | FULL | $1.6443 \pm 0.0029$ | – |

