# OpenReview forum: "On Finetuning Tabular Foundation Models"
_ICLR.cc/2026/Conference — Submitted to ICLR 2026_

### Official Review · Reviewer_r8Z8 · 2025-10-23

**Soundness:** 2
**Presentation:** 2
**Contribution:** 1
**Rating:** 4
**Confidence:** 4

**Summary:**

This paper investigates the finetuning of tabular foundation model TabPFNv2, confirming full finetuning as the optimal strategy , revealing it enhances the model by improving query-key dot product alignment with target similarity and concentrating attention, while noting finetuned TabPFNv2 achieves SOTA on academic datasets but lacks stability on real-world datasets with shifts.

**Strengths:**

1. The paper is clear and easy to understand, with intuitive tables/figures aiding comprehension.

2. It provides a relatively comprehensive investigation into the finetuning of the specific model TabPFNv2, covering strategy evaluation, mechanism analysis and performance comparison.

**Weaknesses:**

1. There are typos that require correction. For example, Line 662 has "concetrated" which should be "concentrated", Line 693 has "intermidiate" which should be "intermediate", and Line 726 has "perfomance" which should be "performance". Though these errors do not affect the understanding of core content, they may reduce the sense of rigor, so a full text check is recommended to fix similar issues.

2. The research contributions are relatively limited. The study only focuses on the finetuning evaluation of TabPFNv2, without exploring the universality of the proposed finetuning strategies on other tabular foundation models or putting forward innovative finetuning theories/methods.

3. The applicable scope of the studied finetuning methods is narrow. They only work for small and medium-sized datasets and lack scalability to larger datasets. Additionally, on TabReD datasets with temporal shifts , finetuned TabPFNv2 shows reduced stability (even performance degradation), failing to adapt to complex real-world scenarios.

4. The baseline comparison is insufficient. The paper does not include mainstream tabular deep learning models like ExcelFormer and RealMLP, nor does it compare with AutoGluon (mentioned in the original TabPFN paper).

**Questions:**

Refer to the Weaknesses.

---

### Official Review · Reviewer_hPwk · 2025-10-24

**Soundness:** 3
**Presentation:** 3
**Contribution:** 2
**Rating:** 4
**Confidence:** 4

**Summary:**

The paper studies single dataset adaptation of TabPFNv2. It compares full fine tuning to several parameter efficient methods (e.g. LoRA), reporting that full fine tuning reaches comparable or better performance with lower wall clock time. The analysis contrasts full fine tuning with training from scratch, examines dataset size effects, and argues that fine tuning improves in context retrieval behavior through better attention to relevant context. Additional experiments on TabRED (temporal shift) suggest TabPFNv2 is less stable than some alternatives. Overall, the empirical message is that simple full fine tuning is a strong and efficient method for adapting TabPFNv2 to a real dataset.

**Strengths:**

- The paper connects fine tuning outcomes to retrieval like behavior via kNN style analyses of representations, which is novel and potentially useful.
- There is a clear claim that full fine tuning is an efficient and competitive choice for TabPFNv2 which is supported through many experiments.
- Comparison to training from scratch and to dataset size scaling is provided that shows fine tuning is superior across many settings.

**Weaknesses:**

- The paper does not provide comparisons to retrieval based adaptation and sub sampling e.g. LocalPFN style retrieve + fine tune, despite being cited and working efficiently with larger datasets and being aligned with the paper's retrieval hypothesis.
- The scope is limited to TabPFNv2, and the conclusions are not validated on other tabular foundation models such as TabICL [1] or TabDPT [2] (both more efficient to fine tune), even though the paper itself notes that findings from TabPFN to TabPFNv2 do not always transfer.
- For Figure 3, it is unclear which attention weights are analyzed in a cell based model and the results are shown for a few datasets and two show minimal effect. There is a potential confound due to context length and attention temperature. Prior work [3] on attention temperature shows that when evaluating on longer context, the attention scores are not as sharp (higher entropy), and therefore a temperature controlled ablation is needed to show that fine tuning differs from simply temperature scaling to adapt the model to a longer context.


[1] Qu, Jingang, et al. "TabICL: A Tabular Foundation Model for In-Context Learning on Large Data." ICML 2025-Forty-Second International Conference on Machine Learning. 2025.

[2] Ma, Junwei, et al. "Tabdpt: Scaling tabular foundation models." arXiv preprint arXiv:2410.18164 (2024).

[3] Veličković, Petar, et al. "Softmax is not Enough (for Sharp Size Generalisation)." Forty-second International Conference on Machine Learning.

**Questions:**

- What is the metric for each dataset in Table 2 (log loss, AUC, MSE, accuracy)?
- For all settings in Table 1, what is the total number of optimization steps until early stop, the average time per step and peak memory usage? Answering this clarifies the reason for speed advantage for full fine tuning.
- What is the representation used for kNN precisely? TabPFNv2 is cell based, how are row level similarities measured?
- Same question as above for Figure 3. How are attention weights across samples (i.e. rows as opposed to cells) calculated?
- Minor text editing suggestions:

    - For Figure 4, the main text should preview the key definition (now in the appendix) where it is stated that the x axis is the sample index sorted by difference of entropy.

    - The paper should explicitly mention overfitting as the cause of failure in the main text (explained in Appendix E), as it is important enough to appear in the main section.

    - Real-TabPFN [4] seems directly relevant and should be cited as it fine tunes TabPFNv2 on real data (although not single dataset).

[4] Garg, Anurag, et al. "Real-tabpfn: Improving tabular foundation models via continued pre-training with real-world data." arXiv preprint arXiv:2507.03971 (2025).

---

### Official Review · Reviewer_WYkp · 2025-10-26

**Soundness:** 4
**Presentation:** 4
**Contribution:** 2
**Rating:** 4
**Confidence:** 5

**Summary:**

This paper investigates different versions of fine-tuning of TabPFN v2. It finds simple fine-tuning being the strongest and most stable version of fine-tuning of TabPFN v2 comparing to other forms of fine-tuning strategies. In addition, this paper examines transformer attention map in details and finds the attention entropy decreases for a fine-tuned model comparing to a non-fine tuned version. Another contribution of this paper is the comparison between different models such as MLP-PLR, MNCA together with the fine-tuned and the non-fine-tuned versions of TabPFN v2.

**Strengths:**

### Originality and Significance
There has been many works examining fine-tuning of tabular foundation models. However, the previous works did not have strong conclusions. The author of this paper offers a slightly stronger conclusion, i.e. "simple full fine-tuning is a strong and stable baseline for TabPFN v2 adaptation" although it is still not very strong. The rigorous comparison with MNCA, MLP-PLR and other models are definitely also original.

I believe the most original and significant contribution comes from the analyses of the attention map and the entropy of the attention scores. The paper finds there is a correlation between the decrease of entropy and the performance improvement. The paper makes analogy of the phenomenon with retrieval, a technique that has been proven very useful in tabular foundation models.

### Quality and Clarity
The paper overall is very polished and very clear with some unclear details (see in weaknesses). The paper chooses established benchmarks and established baselines and the use of confidence intervals makes the results more conclusive.

**Weaknesses:**

### Contribution and Significance
I believe the biggest weakness of the paper is the lack of clear and conclusive contribution. The paper finds full fine-tuning being the strongest and most stable baseline over other versions of fine-tuning, but it does not give any guidance or rule for when to use full fine-tuning vs other forms of fine-tuning or no fine-tuning.

The paper finds there is a correlation between the key/query alignment (or the entropy of the attention scores) and the performance improvement. However, it is still to be investigated what caused both phenomenon. Is the decrease of entropy inevitable after fine-tuning or not? Why are they correlated?

Instead of analyzing different datasets separately (as in figure 4), I believe an aggregated graph of change in entropy vs performance would give a better view of the overall relationship between entropy and performance.

### Choice of Models
Another reason the contribution is limited is because only TabPFN v2 model is used for all experiments for fine-tuning. It would be interesting to see whether TabPFN v1, TabICL, TabDPT also have similar phenomena.

### Minor
Highlights of the best model scores in table 3 would make it much more clear.

**Questions:**

1. The paper compares early on to prior works that uses context optimization PEFT method [line 78]. However, I believe these methods were never compared to later on in the experiments?
2. In table 1, full fine-tuning appears to be faster than other PEFT methods, why is this? Shouldn't PEFT methods be faster?
3. In table 2, are the Pred. lengths indicating the batch size or the sequence length? The term "object" in the caption is a bit confusing because it can mean either one.
4. What happens if we take kNN from different layers instead of the last layer in table 3? Do we see entropy decreasing over the layers?

---

### Official Review · Reviewer_eYTT · 2025-10-31

**Soundness:** 3
**Presentation:** 3
**Contribution:** 1
**Rating:** 2
**Confidence:** 4

**Summary:**

The paper explores fine tuning for tabular foundational models. Specifically, authors explore different fine tuning methods for TabPFNv2 and analyze the impact of fine tuning on attention.

**Strengths:**

The paper is well written and explores an important aspect of foundation models. Analysis is anchored in real data and clearly shows that fine tuning is beneficial for model performance. Extensive empirical evaluation is conducted with diverse datasets and leading baselines.

**Weaknesses:**

I think the paper lacks novelty and provides fairly obvious conclusions that 1) fine tuning helps 2) full fine tuning is better than PEFT 3) fine tuning makes attention better (more peaked) at selecting the relevant information from the input context. All of these conclusion have been shown multiple times before on LLMs and other foundation models. So I think more is need to differentiate on novelty.

**Questions:**

None

---

### Meta-Review · Area_Chair_xyAJ · 2025-12-23

**Summary:**

This paper presents an empirical study of fine-tuning strategies for the tabular foundation model TabPFNv2, analyzing full fine-tuning versus parameter-efficient methods and examining changes in attention behavior. While reviewers generally found the paper to be clearly written and empirically thorough, the dominant concern across reviews was the limited novelty and contribution. The main conclusions, that full fine-tuning is often the strongest and most stable adaptation strategy and that fine-tuning sharpens attention, were viewed as largely expected and previously observed in other foundation model settings. Additional concerns included the narrow scope restricted to a single model (TabPFNv2), lack of validation on other tabular foundation models, insufficient comparison to relevant retrieval-based or hybrid adaptation methods, unclear causal interpretation of the attention entropy analysis, and limited relevance to more challenging real-world or large-scale settings. Taken together, reviewers agreed that while the empirical findings are solid, the work does not meet the bar for ICLR.

**Reviewer Concerns:**

Several concerns were consistently raised and remain outstanding. Most notably, the paper’s contribution was viewed as incremental, with conclusions that largely confirm known behavior of fine-tuning in foundation models, without offering new methods, theory, or actionable guidance on when different adaptation strategies should be preferred. The experimental scope is limited to TabPFNv2, leaving open whether the findings generalize to other prominent tabular foundation models (e.g., TabICL, TabDPT). Reviewers also noted missing or insufficient comparisons to retrieval-based adaptation approaches and mainstream tabular baselines, as well as unresolved questions about the causal relationship between attention entropy and performance, including potential confounds such as context length and attention temperature. Minor issues related to clarity, typos, and reporting details were identified but are not central to the decision. As there was no author rebuttal, none of these concerns were substantively addressed during the discussion phase.

**Reviewer Scores:**

Reviewer eYTT was confident in a reject recommendation due to lack of novelty and would be unlikely to change their score. Reviewer WYkp rated the paper marginally below threshold and appreciated the empirical rigor and attention analysis, but emphasized limited contribution; without rebuttal or broader validation, their score would likely remain unchanged. Reviewer hPwk similarly viewed the paper as solid but incremental, with significant missing comparisons and scope limitations; their score would likely not increase in the absence of responses. Reviewer r8Z8 highlighted narrow applicability, limited baselines, and instability under dataset shifts; without rebuttal, their assessment would also remain at marginally below threshold. Overall, no reviewer indicated issues that were resolved or clarified sufficiently to justify a higher score.

---

### Decision · Program_Chairs · 2026-01-26

Reject